# Achieving Linear Speedup with Partial Worker Participation in Non-IID Federated Learning

**Haibo Yang, Minghong Fang, and Jia Liu**
Department of Electrical and Computer Engineering
The Ohio State University
Columbus, OH 43210 USA
`{yang.5952, fang.841, liu.1736}@osu.edu`

## ABSTRACT

Federated learning (FL) is a distributed machine learning architecture that leverages a large number of workers to jointly learn a model with decentralized data. FL has received increasing attention in recent years thanks to its data privacy protection, communication efficiency and a linear speedup for convergence in training (i.e., convergence performance increases linearly with respect to the number of workers). However, existing studies on linear speedup for convergence are only limited to the assumptions of i.i.d. datasets across workers and/or full worker participation, both of which rarely hold in practice. So far, it remains an open question whether or not the linear speedup for convergence is achievable under non-i.i.d. datasets with partial worker participation in FL. In this paper, we show that the answer is affirmative. Specifically, we show that the federated averaging (FedAvg) algorithm (with two-sided learning rates) on non-i.i.d. datasets in non-convex settings achieves a convergence rate $\mathcal{O}(\frac{1}{\sqrt{mKT}} + \frac{1}{T})$ for full worker participation and a convergence rate $\mathcal{O}(\frac{\sqrt{K}}{\sqrt{nT}} + \frac{1}{T})$ for partial worker participation, where $K$ is the number of local steps, $T$ is the number of total communication rounds, $m$ is the total worker number and $n$ is the worker number in one communication round if for partial worker participation. Our results also reveal that the local steps in FL could help the convergence and show that the maximum number of local steps can be improved to $T/m$ in full worker participation. We conduct extensive experiments on MNIST and CIFAR-10 to verify our theoretical results.

## 1 INTRODUCTION

Federated Learning (FL) is a distributed machine learning paradigm that leverages a large number of workers to collaboratively learn a model with decentralized data under the coordination of a centralized server. Formally, the goal of FL is to solve an optimization problem, which can be decomposed as:

$$\min_{x \in \mathbb{R}^d} f(x) := \frac{1}{m} \sum_{i=1}^{m} F_i(x),$$

where $F_i(x) \triangleq \mathbb{E}_{\xi_i \sim D_i}[F_i(x, \xi_i)]$ is the local (non-convex) loss function associated with a local data distribution $D_i$ and $m$ is the number of workers. FL allows a large number of workers (such as edge devices) to participate flexibly without sharing data, which helps protect data privacy. However, it also introduces two unique challenges unseen in traditional distributed learning algorithms that are used typically for large data centers:

- **Non-independent-identically-distributed (non-i.i.d.) datasets across workers** (data heterogeneity): In conventional distributed learning in data centers, the distribution for each worker's local dataset can usually be assumed to be i.i.d., i.e., $D_i = D, \forall i \in \{1, ..., m\}$. Unfortunately, this assumption rarely holds for FL since data are generated locally at the workers based on their circumstances, i.e., $D_i \neq D_j$, for $i \neq j$. It will be seen later that the non-i.i.d assumption imposes significant challenges in algorithm design for FL and their performance analysis.

- **Time-varying partial worker participation** (systems non-stationarity): With the flexibility for workers' participation in many scenarios (particularly in mobile edge computing), workers may randomly join or leave the FL system at will, thus rendering the active worker set stochastic and time-varying across communication rounds. Hence, it is often infeasible to wait for all workers' responses as in traditional distributed learning, since inactive workers or stragglers will significantly slow down the whole training process. As a result, only a subset of the workers may be chosen by the server in each communication round, i.e., partial worker participation.

In recent years, the Federated Averaging method (FedAvg) and its variants (McMahan et al., 2016; Li et al., 2018; Hsu et al., 2019; Karimireddy et al., 2019; Wang et al., 2019a) have emerged as a prevailing approach for FL. Similar to the traditional distributed learning, FedAvg leverages local computation at each worker and employs a centralized parameter server to aggregate and update the model parameters. The unique feature of FedAvg is that each worker runs *multiple local stochastic gradient descent (SGD) steps* rather than just one step as in traditional distributed learning between two consecutive communication rounds. For i.i.d. datasets and the full worker participation setting, Stich (2018) and Yu et al. (2019b) proposed two variants of FedAvg that achieve a convergence rate of $\mathcal{O}(\frac{mK}{T} + \frac{1}{\sqrt{mKT}})$ with a bounded gradient assumption for both strongly convex and non-convex problems, where $m$ is the number of workers, $K$ is the local update steps, and $T$ is the total communication rounds. Wang & Joshi (2018) and Stich & Karimireddy (2019) further proposed improved FedAvg algorithms to achieve an $\mathcal{O}(\frac{m}{T} + \frac{1}{\sqrt{mKT}})$ convergence rate without bounded gradient assumption. Notably, for a sufficiently large $T$, the above rates become $\mathcal{O}(\frac{1}{\sqrt{mKT}})$[1], which implies a **linear speedup** with respect to the number of workers.[2] This linear speedup is highly desirable for an FL algorithm because the algorithm is able to effectively leverage the massive parallelism in a large FL system. However, with non-i.i.d. datasets and partial worker participation in FL, a fundamental open question arises: *Can we still achieve the same linear speedup for convergence, i.e., $\mathcal{O}(\frac{1}{\sqrt{mKT}})$, with non-i.i.d. datasets and under either full or partial worker participation?*

In this paper, we show the answer to the above question is affirmative. Specifically, we show that a *generalized FedAvg with two-sided learning rates* achieves linear convergence speedup with non-i.i.d. datasets and under full/partial worker participation. We highlight our contributions as follows:

- For non-convex problems, we show that the convergence rate of the FedAvg algorithm on non-i.i.d. dataset are $\mathcal{O}(\frac{1}{\sqrt{mKT}} + \frac{1}{T})$ and $\mathcal{O}(\frac{\sqrt{K}}{\sqrt{nT}} + \frac{1}{T})$ for full and partial worker participation, respectively, where $n$ is the size of the partially participating worker set. This indicates that our proposed algorithm achieves a linear speedup for convergence rate for a sufficiently large $T$. When reduced to the i.i.d. case, our convergence rate is $\mathcal{O}(\frac{1}{TK} + \frac{1}{\sqrt{mKT}})$, which is also better than previous works. We summarize the convergence rate comparisons for both i.i.d. and non-i.i.d. cases in Table 1. It is worth noting that our proof does not require the bounded gradient assumption. We note that the SCAFFOLD algorithm (Karimireddy et al., 2019) also achieves the linear speedup but extra variance reduction operations are required, which lead to higher communication costs and implementation complexity. By contrast, we do not have such extra requirements in this paper.

- In order to achieve a linear speedup, i.e., a convergence rate $\mathcal{O}(\frac{1}{\sqrt{mKT}})$, we show that the number of local updates $K$ can be as large as $T/m$, which improves the $T^{1/3}/m$ result previously shown in Yu et al. (2019a) and Karimireddy et al. (2019). As shown later in the communication complexity comparison in Table 1, a larger number of local steps implies relatively fewer communication rounds, thus less communication overhead. Interestingly, our results also indicate that the number of local updates $K$ does not hurt but rather help the convergence with a proper learning rates choice in full worker participation. This overcomes the limitation as suggested in Li et al. (2019b) that local SGD steps might slow down the convergence ($\mathcal{O}(\frac{K}{T})$ for strongly convex case). This result also reveals new insights on the relationship between the number of local steps and learning rate.

---

[1]This rate also matches the convergence rate order of parallel SGD in conventional distributed learning.

[2]To attain $\epsilon$ accuracy for an algorithm, it needs to take $\mathcal{O}(\frac{1}{\epsilon^2})$ steps with a convergence rate $\mathcal{O}(\frac{1}{\sqrt{T}})$, while needing $\mathcal{O}(\frac{1}{m\epsilon^2})$ steps if the convergence rate is $\mathcal{O}(\frac{1}{\sqrt{mT}})$ (the hidden constant in Big-O is the same). In this sense, one achieves a *linear speedup* with respect to the number of workers.

Table 1: Convergence rates of optimization methods for FL.

| Dataset | Algorithm[6] | Convexity[7] | Partial Worker | Convergence Rate | Communication complexity |
|---------|-------------|--------------|----------------|------------------|--------------------------|
| IID | Stich1 | SC | × | $\mathcal{O}(\frac{mK}{T} + \frac{1}{\sqrt{mKT}})$ | $\mathcal{O}(\frac{mK}{\epsilon} + \frac{1}{mK\epsilon^2})$ |
| | Yu1 | NC | × | $\mathcal{O}(\frac{mK}{T} + \frac{1}{\sqrt{mKT}})$ | $\mathcal{O}(\frac{mK}{\epsilon} + \frac{1}{mK\epsilon^2})$ |
| | Wang | NC | × | $\mathcal{O}(\frac{m}{T} + \frac{1}{\sqrt{mKT}})$ | $\mathcal{O}(\frac{m}{\epsilon} + \frac{1}{mK\epsilon^2})$ |
| | Stich2 | NC | × | $\mathcal{O}(\frac{m}{T} + \frac{1}{\sqrt{mKT}})$ | $\mathcal{O}(\frac{m}{\epsilon} + \frac{1}{mK\epsilon^2})$ |
| | **This paper** | **NC** | ✓ | $\mathcal{O}(\frac{1}{TK} + \frac{1}{\sqrt{mKT}})$ | $\mathcal{O}(\frac{1}{K\epsilon} + \frac{1}{mK\epsilon^2})$ |
| NON-IID | Khaled [1] | C | × | $\mathcal{O}(\frac{m}{T} + \frac{1}{\sqrt{mT}})$ | $\mathcal{O}(\frac{m}{\epsilon} + \frac{1}{mK\epsilon^2})$ |
| | Yu2[2] | NC | × | $\mathcal{O}(\frac{m}{TK} + \frac{1}{\sqrt{mKT}})$ | $\mathcal{O}(\frac{m}{K\epsilon} + \frac{1}{mK\epsilon^2})$ |
| | Li | SC | ✓ | $\mathcal{O}(\frac{K}{T})$ | $\mathcal{O}(\frac{K}{\epsilon})$ |
| | Karimireddy [3] | NC | ✓ | $\mathcal{O}(\frac{1}{T^{2/3}} + \frac{M}{\sqrt{SKT}})$ | $\mathcal{O}(\frac{1}{\epsilon^{3/2}} + \frac{M}{SK\epsilon^2})$ |
| | Karimireddy [4] | NC | ✓ | $\mathcal{O}(\frac{1}{T} + \frac{1}{\sqrt{mKT}})$ | $\mathcal{O}(\frac{1}{\epsilon} + \frac{1}{mK\epsilon^2})$ |
| | **This paper**[5] | **NC** | ✔ | $\mathcal{O}(\frac{1}{T} + \frac{1}{\sqrt{mKT}})$ | $\mathcal{O}(\frac{1}{\epsilon} + \frac{1}{mK\epsilon^2})$ |

[1] Full gradients are used for each worker.

[2] Local momentum is used at each worker.

[3] A FedAvg algorithm with two-sided learning rates. $M^2 = \mathcal{O}(1) + \mathcal{O}(KS(1 - \frac{S}{m}))$. $S = m$ ($S = n$) for full (partial) worker participation.

[4] The SCAFFOLD algorithm in Karimireddy et al. (2019) for non-convex case.

[5] The convergence rate becomes $\mathcal{O}(\frac{1}{T} + \frac{\sqrt{K}}{\sqrt{nT}})$ under partial worker participation.

[6] Shorthand notation for references: Stich1 := Stich (2018), Yu2 := Yu et al. (2019b), Wang:= Wang & Joshi (2018), Stich2:= Stich & Karimireddy (2019); Khaled:= Khaled et al. (2019b), Yu2:=Yu et al. (2019a), Li:= Li et al. (2019b), and Karimireddy:= Karimireddy et al. (2019).

[7] Shorthand notation for convexity: SC: Strongly Convex, C: Convex, and NC: Non-Convex.

**Notation.** In this paper, we let $m$ be the total number of workers and $S_t$ be the set of active workers for the $t$-th communication round with size $|S_t| = n$ for some $n \in (0, m]$. [3] We use $K$ to denote the number of local steps per communication round at each worker. We let $T$ be the number of total communication rounds. In addition, we use boldface to denote matrices/vectors. We let $[\cdot]_{t,k}^i$ represent the parameter of $k$-th local step in the $i$-th worker after the $t$-th communication. We use $\|\cdot\|_2$ to denote the $\ell^2$-norm. For a natural number $m$, we use $[m]$ to represent the set $\{1, \cdots, m\}$.

The rest of the paper is organized as follows. In Section 2, we review the literature to put our work in comparative perspectives. Section 3 presents the convergence analysis for our proposed algorithm. Section 4 discusses the implication of the convergence rate analysis. Section 5 presents numerical results and Section 6 concludes this paper. Due to space limitation, the details of all proofs and some experiments are provided in the supplementary material.

## 2 RELATED WORK

The federated averaging (FedAvg) algorithm was first proposed by McMahan et al. (2016) for FL as a heuristic to improve communication efficiency and data privacy. Since then, this work has sparked many follow-ups that focus on FL with i.i.d. datasets and full worker participation (also known as LocalSGD (Stich, 2018; Yu et al., 2019b; Wang & Joshi, 2018; Stich & Karimireddy, 2019; Lin et al., 2018; Khaled et al., 2019a; Zhou & Cong, 2017)). Under these two assumptions, most of the theoretical works can achieve a linear speedup for convergence, i.e., $\mathcal{O}(\frac{1}{\sqrt{mKT}})$ for a sufficiently large $T$, matching the rate of the parallel SGD. In addition, LocalSGD is empirically shown to be communication-efficient and enjoys better generalization performance (Lin et al., 2018). For a comprehensive introduction to FL, we refer readers to Li et al. (2019a) and Kairouz et al. (2019).

---

[3]For simplicity and ease of presentation in this paper, we let $|S_t| = n$. We note that this is not a restrictive condition and our proofs and results still hold for $|S_t| \geq n$, which can be easily satisfied in practice.

---

**Algorithm 1** A Generalized FedAvg Algorithm with Two-Sided Learning Rates.

---

Initialize $\mathbf{x}_0$
**for** $t = 0, \cdots, T - 1$ **do**
    The server samples a subset $S_t$ of workers with $|S_t| = n$.
    **for** each worker $i \in S_t$ in parallel **do**
        $\mathbf{x}_{t,0}^i = \mathbf{x}_t$
        **for** $k = 0, \cdots, K - 1$ **do**
            Compute an unbiased estimate $\mathbf{g}_{t,k}^i = \nabla F_i(\mathbf{x}_{t,k}^i, \xi_{t,k}^i)$ of $\nabla F_i(\mathbf{x}_{t,k}^i)$.
            Local worker update: $\mathbf{x}_{t,k+1}^i = \mathbf{x}_{t,k}^i - \eta_L \mathbf{g}_{t,k}^i$.
        **end for**
        Let $\Delta_t^i = \mathbf{x}_{t,K}^i - \mathbf{x}_{t,0}^i = -\eta_L \sum_{k=0}^{K-1} \mathbf{g}_{t,k}^i$. Send $\Delta_t^i$ to the server.
    **end for**
    At Server:
        Receive $\Delta_t^i, i \in S$.
        Let $\Delta_t = \frac{1}{|S|} \sum_{i \in S} \Delta_t^i$.
        Server Update: $\mathbf{x}_{t+1} = \mathbf{x}_t + \eta \Delta_t$.
        Broadcasting $\mathbf{x}_{t+1}$ to workers.
**end for**

---

For non-i.i.d. datasets, many works (Sattler et al., 2019; Zhao et al., 2018; Li et al., 2018; Wang et al., 2019a; Karimireddy et al., 2019; Huang et al., 2018; Jeong et al., 2018) heuristically demonstrated the performance of FedAvg and its variants. On convergence rate with full worker participation, many works (Stich et al., 2018; Yu et al., 2019a; Wang & Joshi, 2018; Karimireddy et al., 2019; Reddi et al., 2020) can achieve linear speedup, but their convergence rate bounds could be improved as shown in this paper. On convergence rate with partial worker participation, Li et al. (2019b) showed that the original FedAvg can achieve $\mathcal{O}(K/T)$ for strongly convex functions, which suggests that local SGD steps slow down the convergence in the original FedAvg. Karimireddy et al. (2019) analyzed a generalized FedAvg with two-sided learning rates under strongly convex, convex and non-convex cases. However, as shown in Table 1, none of them indicates that linear speedup is achievable with non-i.i.d. datasets under partial worker participation. Note that the SCAFFOLD algorithm (Karimireddy et al., 2019) can achieve linear speedup but extra variance reduction operations are required, which lead to higher communication costs and implementation complexity. In this paper, we show that this linear speedup can be achieved *without* any extra requirements. For more detailed comparisons and other algorithmic variants in FL and decentralized settings, we refer readers to Kairouz et al. (2019).

## 3 LINEAR SPEEDUP OF THE GENERALIZED FEDAVG WITH TWO-SIDED LEARNING RATES FOR NON-IID DATASETS

In this paper, we consider a FedAvg algorithm with two-sided learning rates as shown in Algorithm 1, which is generalized from previous works (Karimireddy et al., 2019; Reddi et al., 2020). Here, workers perform multiple SGD steps using a worker optimizer to minimize the local loss on its own dataset, while the server aggregates and updates the global model using another gradient-based server optimizer based on the returned parameters. Specifically, between two consecutive communication rounds, each worker performs $K$ SGD steps with the worker's local learning rate $\eta_L$. We assume an unbiased estimator in each step, which is denoted by $\mathbf{g}_{t,k}^i = \nabla F_i(\mathbf{x}_{t,k}^i, \xi_{t,k}^i)$, where $\xi_{t,k}^i$ is a random *local* data sample for $k$-th steps after $t$-th communication round at worker $i$. Then, each worker sends the accumulative parameter difference $\Delta_t^i$ to the server. On the server side, the server aggregates all available $\Delta_t^i$-values and updates the model parameters with a *global* learning rate $\eta$. The FedAvg algorithm with two-sided learning rates provides a natural way to decouple the learning of workers and server, thus utilizing different learning rate schedules for workers and the server. The original FedAvg can be viewed as a special case of this framework with server-side learning rate being one.

In what follows, we show that a linear speedup for convergence is achievable by the generalized FedAvg for non-convex functions on non-i.i.d. datasets. We first state our assumptions as follows.

**Assumption 1.** *(L-Lipschitz Continuous Gradient) There exists a constant $L > 0$, such that $\|\nabla F_i(\mathbf{x}) - \nabla F_i(\mathbf{y})\| \leq L\|\mathbf{x} - \mathbf{y}\|, \forall \mathbf{x}, \mathbf{y} \in \mathbb{R}^d, and\ i \in [m]$.*

**Assumption 2.** *(Unbiased Local Gradient Estimator) Let $\xi_t^i$ be a random local data sample in the $t$-th step at the $i$-th worker. The local gradient estimator is unbiased, i.e., $\mathbb{E}[\nabla F_i(\mathbf{x}_t, \xi_t^i)] = \nabla F_i(\mathbf{x}_t)$, $\forall i \in [m]$, where the expectation is over all local datasets samples.*

**Assumption 3.** *(Bounded Local and Global Variance) There exist two constants $\sigma_L > 0$ and $\sigma_G > 0$, such that the variance of each local gradient estimator is bounded by $\mathbb{E}[\|\nabla F_i(\mathbf{x}_t, \xi_t^i) - \nabla F_i(\mathbf{x}_t)\|^2] \leq \sigma_L^2, \forall i \in [m]$, and the global variability of the local gradient of the cost function is bounded by $\|\nabla F_i(\mathbf{x}_t) - \nabla f(\mathbf{x}_t)\|^2 \leq \sigma_G^2, \forall i \in [m], \forall t$.*

The first two assumptions are standard in non-convex optimization (Ghadimi & Lan, 2013; Bottou et al., 2018). For Assumption 3, the bounded local variance is also a standard assumption. We use a universal bound $\sigma_G$ to quantify the heterogeneity of the non-i.i.d. datasets among different workers. In particular, $\sigma_G = 0$ corresponds to i.i.d. datasets. This assumption is also used in other works for FL under non-i.i.d. datasets (Reddi et al., 2020; Yu et al., 2019b; Wang et al., 2019b) as well as in decentralized optimization (Kairouz et al., 2019). It is worth noting that we do *not* require a bounded gradient assumption, which is often assumed in FL optimization analysis.

### 3.1 CONVERGENCE ANALYSIS FOR FULL WORKER PARTICIPATION

In this subsection, we first analyze the convergence rate of the generalized FedAvg with two-sided learning rates under full worker participation, for which we have the following result:

**Theorem 1.** *Let constant local and global learning rates $\eta_L$ and $\eta$ be chosen as such that $\eta_L \leq \frac{1}{8LK}$ and $\eta\eta_L \leq \frac{1}{KL}$. Under Assumptions 1–3 and with full worker participation, the sequence of outputs $\{\mathbf{x}_k\}$ generated by Algorithm 1 satisfies:*

$$\min_{t \in [T]} \mathbb{E}[\|\nabla f(\mathbf{x}_t)\|_2^2] \leq \frac{f_0 - f_*}{c\eta\eta_L KT} + \Phi,$$

*where $\Phi \triangleq \frac{1}{c}[\frac{L\eta\eta_L}{2m}\sigma_L^2 + \frac{5K\eta_L^2 L^2}{2}(\sigma_L^2 + 6K\sigma_G^2)]$, $c$ is a constant, $f_0 \triangleq f(\mathbf{x}_0)$, $f_* \triangleq f(\mathbf{x}_*)$ and the expectation is over the local dataset samples among workers.*

**Remark 1.** The convergence bound contains two parts: a vanishing term $\frac{f_0 - f_*}{c\eta\eta_L KT}$ as $T$ increases and a constant term $\Phi$ whose size depends on the problem instance parameters and is independent of $T$. The vanishing term's decay rate matches that of the typical SGD methods.

**Remark 2.** The first part of $\Phi$ (i.e., $\frac{L\eta\eta_L}{2m}\sigma_L^2$) is due to the local stochastic gradients at each worker, which shrinks at rate $\frac{1}{m}$ as $m$ increases. The cumulative variance of the $K$ local steps contributes to the second term in $\Phi$ (i.e., $\frac{5K\eta_L^2 L^2}{2}(\sigma_L^2 + 6K\sigma_G^2)$), which is independent of $m$ and largely affected by the data heterogeneity. To make the second part small, an inverse relationship between the local learning rate and local steps should be satisfied, i.e., $\eta_L = \mathcal{O}(\frac{1}{K})$. Specifically, note that the global and local variances are quadratically and linearly amplified by $K$. This requires a sufficiently small $\eta_L$ to offset the variance between two successive communication rounds to make the second term in $\Phi$ small. This is consistent with the observation in strongly convex FL that a decaying learning rate is needed for FL to converge under non-i.i.d. datasets even if full gradients used in each worker (Li et al., 2019b). However, we note that our explicit inverse relationship between $\eta_L$ and $K$ in the above is new. Intuitively, the $K$ local steps with a sufficiently small $\eta_L$ can be viewed as one SGD step with a large learning rate.

With Theorem 1, we immediately have the following convergence rate for the generalized FedAvg algorithm with a proper choice of two-sided learning rates:

**Corollary 1.** *Let $\eta_L = \frac{1}{\sqrt{T}KL}$ and $\eta = \sqrt{Km}$. The convergence rate of the generalized FedAvg algorithm under full worker participation is $\min_{t \in [T]} \mathbb{E}[\|\nabla f(\mathbf{x}_t)\|_2^2] = \mathcal{O}\left(\frac{1}{\sqrt{mKT}} + \frac{1}{T}\right)$.*

**Remark 3.** The generalized FedAvg algorithm with two-sided learning rates can achieve a linear speedup for non-i.i.d. datasets, i.e., a $\mathcal{O}(\frac{1}{\sqrt{mKT}})$ convergence rate as long as $T \geq mK$. Although many works have achieved this convergence rate asymptotically, we improve the maximum number

of local steps $K$ to $T/m$, which is significantly better than the state-of-art bounds such as $T^{1/3}/m$ shown in (Karimireddy et al., 2019; Yu et al., 2019a; Kairouz et al., 2019). Note that a larger number of local steps implies relatively fewer communication rounds, thus less communication overhead. See also the communication complexity comparison in Table 1. For example, when $T = 10^6$ and $m = 100$ (as used in (Kairouz et al., 2019)), the local steps in our algorithm is $K \leq T/m = 10^4$. However, $K \leq \frac{T^{1/3}}{m} = 1$ means that no extra local steps can be taken to reduce communication costs.

**Remark 4.** When degenerated to the i.i.d. case ($\sigma_G = 0$), the convergence rate becomes $\mathcal{O}(\frac{1}{TK} + \frac{1}{\sqrt{mKT}})$, which has a better first term in the bound compared with previous work as shown in Table 1.

### 3.2 CONVERGENCE ANALYSIS FOR PARTIAL WORKER PARTICIPATION

Partial worker participation in each communication round may be more practical than full worker participation due to many physical limitations of FL in practice (e.g., excessive delays because of too many devices to poll, malfunctioning devices, etc.). Partial worker participation can also accelerate the training by neglecting stragglers. We consider two sampling strategies proposed by Li et al. (2018) and Li et al. (2019b). Let $S_t$ be the participating worker index set at communication round $t$ with $|S_t| = n, \forall t$, for some $n \in (0, m]$. $S_t$ is randomly and independently selected either with replacement (Strategy 1) or without replacement (Strategy 2) sequentially according to the sampling probabilities $p_i, \forall i \in [m]$. For each member in $S_t$, we pick a worker from the entire set $[m]$ uniformly at random with probability $p_i = \frac{1}{m}, \forall i \in [m]$. That is, selection likelihood for anyone worker $i \in S_t$ is $p = \frac{n}{m}$. Then we have the following results:

**Theorem 2.** *Under Assumptions 1–3 with partial worker participation, the sequence of outputs $\{\mathbf{x}_k\}$ generated by Algorithm 1 with constant learning rates $\eta$ and $\eta_L$ satisfies:*

$$\min_{t \in [T]} \mathbb{E}[\|\nabla f(\mathbf{x}_t)\|_2^2] \leq \frac{f_0 - f_*}{c\eta\eta_L KT} + \Phi,$$

*where $f_0 = f(\mathbf{x}_0)$, $f_* = f(\mathbf{x}_*)$, and the expectation is over the local dataset samples among workers.*

*For sampling Strategy 1, let $\eta$ and $\eta_L$ be chosen as such that $\eta_L \leq \frac{1}{8LK}$, $\eta\eta_L KL < \frac{n-1}{n}$ and $30K^2\eta_L^2 L^2 - \frac{L\eta\eta_L}{n}(90K^3L^2\eta_L^2 + 3K) < 1$. It then holds that:*

$$\Phi \triangleq \frac{1}{c}\left[\frac{L\eta\eta_L}{2n}\sigma_L^2 + \frac{3LK\eta\eta_L}{2n}\sigma_G^2 + (\frac{5K\eta_L^2 L^2}{2} + \frac{15K^2\eta\eta_L^3 L^3}{2n})(\sigma_L^2 + 6K\sigma_G^2)\right].$$

*For sampling Strategy 2, let $\eta$ and $\eta_L$ be chosen as such that $\eta_L \leq \frac{1}{8LK}$, $\eta\eta_L KL \leq \frac{n(m-1)}{m(n-1)}$ and $10K^2\eta_L^2 L^2 - L\eta\eta_L \frac{m-n}{n(m-1)}(90K^3\eta_L^2 L^2 + 3K) < 1$. It then holds that:*

$$\Phi \triangleq \frac{1}{c}\left[\frac{L\eta\eta_L}{2n}\sigma_L^2 + 3LK\eta\eta_L\frac{m-n}{2n(m-1)}\sigma_G^2 + \left(\frac{5K\eta_L^2 L^2}{2} + 15K^2\eta\eta_L^3 L^3\frac{m-n}{2n(m-1)}\right)(\sigma_L^2 + 6K\sigma_G^2)\right].$$

From Theorem 2, we immediately have the following convergence rate for the generalized FedAvg algorithm with a proper choice of two-sided learning rates:

**Corollary 2.** *Let $\eta_L = \frac{1}{\sqrt{TKL}}$ and $\eta = \sqrt{Kn}$. The convergence rate of the generalized FedAvg algorithm under partial worker participation and both sampling strategies are:*

$$\min_{t \in [T]} \mathbb{E}\|\nabla f(\mathbf{x}_t)\|_2^2 \leq \mathcal{O}\left(\frac{\sqrt{K}}{\sqrt{nT}} + \frac{1}{T}\right).$$

**Remark 5.** The convergence rate bound for partial worker participation has the same structure but with a larger variance term. This implies that the partial worker participation through the uniform sampling does not result in fundamental changes in convergence (in order sense) except for an amplified variance due to fewer workers participating and random sampling. The intuition is that the uniform sampling (with/without replacement) for worker selection yields a good approximation of the entire worker distribution in expectation, which reduces the risk of distribution deviation due to the partial worker participation. As shown in Section 5, the distribution deviation due to fewer worker participation could render the training unstable, especially in highly non-i.i.d. cases.

**Remark 6.** The generalized FedAvg with partial worker participation under non-i.i.d. datasets can still achieve a linear speedup $\mathcal{O}(\frac{\sqrt{K}}{\sqrt{nT}})$ with proper learning rate settings as shown in Corollary 2. In addition, when degenerated to i.i.d. case ($\sigma_G = 0$), the convergence rate becomes $\mathcal{O}(\frac{1}{TK} + \frac{1}{\sqrt{nKT}})$.

**Remark 7.** Here, we let $|S_t| = n$ only for ease of presentation and better readability. We note that this is not a restrictive condition. We can show that $|S_t| = n$ can be relaxed to $|S_t| \geq n, \forall t \in [T]$ and the same convergence rate still holds. In fact, our full proof in Appendix A.2 is for $|S_t| \geq n$.

## 4 Discussion

In light of above results, in what follows, we discuss several insights from the convergence analysis:

**Convergence Rate:** We show that the generalized FedAvg algorithm with two-sided learning rates can achieve a linear speedup, i.e., an $\mathcal{O}(\frac{1}{\sqrt{mKT}})$ convergence rate with a proper choice of hyper-parameters. Thus, it works well in large FL systems, where massive parallelism can be leveraged to accelerate training. The key challenge in convergence analysis stems from the different local loss functions (also called "model drift" in the literature) among workers due to the non-i.i.d. datasets and local steps. As shown above, we obtain a convergence bound for the generalized FedAvg method containing a vanishing term and a constant term (the constant term is similar to that of SGD). In contrast, the constant term in SGD is only due to the local variance. Note that, similar to SGD, the iterations do not diminish the constant term. The local variance $\sigma_L^2$ (randomness of stochastic gradients), global variability $\sigma_G^2$ (non-i.i.d. datasets), and the number of local steps $K$ (amplification factor) all contribute to the constant term, but the total global variability in $K$ local steps dominates the term. When the local learning rate $\eta_L$ is set to an inverse relationship with respect to the number of local steps $K$, the constant term is controllable. An intuitive explanation is that the $K$ small local steps can be approximately viewed as one large step in conventional SGD. So this speedup and the more allowed local steps can be largely attributed to the two-sided learning rates setting.

**Number of Local Steps:** Besides the result that the maximum number of local steps is improved to $K \leq T/m$, we also show that the local steps could help the convergence with the proper hyper-parameter choices, which supports previous numerical results (McMahan et al., 2016; Stich, 2018; Lin et al., 2018) and is verified in different models with different non-i.i.d. degree datasets in Section 5. However, there are other results showing the local steps slow down the convergence (Li et al., 2019b). We believe that whether local steps help or hurt the convergence in FL worths further investigations.

**Number of Workers:** We show that the convergence rate improves substantially as the the number of workers in each communication round increases. This is consistent with the results for i.i.d. cases in Stich (2018). For i.i.d. datasets, more workers means more data samples and thus less variance and better performance. For non-i.i.d. datasets, having more workers implies that the distribution of the sampled workers is a better approximation for the distribution of all workers. This is also empirically observed in Section 5. On the other hand, the sampling strategy plays an important role in non-i.i.d. case as well. Here, we adopt the uniform sampling (with/without replacement) to enlist workers to participate in FL. Intuitively, the distribution of the sampled workers' collective datasets under uniform sampling yields a good approximation of the overall data distribution in expectation.

Note that, in this paper, we assume that every worker is available to participate once being enlisted. However, this may not always be feasible. In practice, the workers need to be in certain states in order to be able to participate in FL (e.g., in charging or idle states, etc. (Eichner et al., 2019)). Therefore, care must be taken in sampling and enlisting workers in practice. We believe that the joint design of sampling schemes and the generalized FedAvg algorithm will have a significant impact on the convergence, which needs further investigations.

## 5 Numerical Results

We perform extensive experiments to verify our theoretical results. We use three models: logistic regression (LR), a fully-connected neural network with two hidden layers (2NN) and a convolution neural network (CNN) with the non-i.i.d. version of MNIST (LeCun et al., 1998) and one ResNet model with CIFAR-10 (Krizhevsky et al., 2009). Due to space limitation, we relegate some experimental results in the supplementary material.

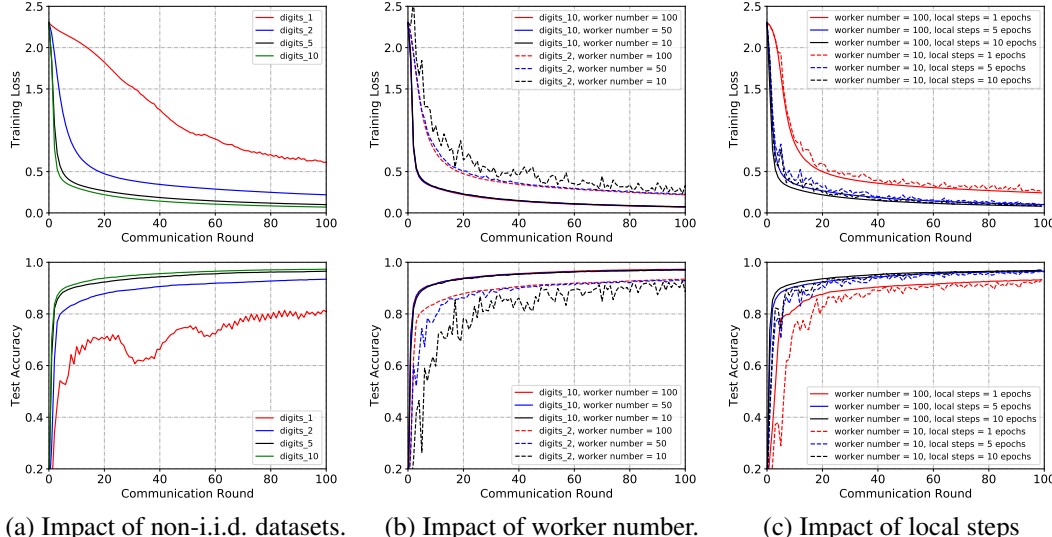

(a) Impact of non-i.i.d. datasets.  (b) Impact of worker number.  (c) Impact of local steps

Figure 1: Training loss (top) and test accuracy (bottom) for the 2NN model with hyper-parameters setting: local learning rate 0.1, global learning rate 1.0: (a) worker number 100, local steps 5 epochs; (b) local steps 5 epochs; (c) 5 digits in each worker's dataset.

In this section, we elaborate the results under non-i.i.d. MNIST datasets for the 2NN. We distribute the MNIST dataset among $m = 100$ workers randomly and evenly in a digit-based manner such that the local dataset for each worker contains only a certain class of digits. The number of digits in each worker's dataset represents the non-i.i.d. degree. For $digits\_10$, each worker has training/testing samples with ten digits from 0 to 9, which is essentially an i.i.d. case. For $digits\_1$, each worker has samples only associated with one digit, which leads to highly non-i.i.d. datasets among workers. For partial worker participation, we set the number of workers $n = 10$ in each communication round.

**Impact of non-i.i.d. datasets:** As shown in Figure 1(a), for the 2NN model with full worker participation, the top-row figures are for training loss versus communication round and the bottom-row are for test accuracy versus communication round. We can see that the generalized FedAvg algorithm converges under non-i.i.d. datasets with a proper learning rate choice in both cases. For five digits ($digits\_5$) in each worker's dataset with full (partial) worker participation in Figure 1(a), the generalized FedAvg algorithm achieves a convergence speed comparable to that of the i.i.d. case ($digits\_10$). Another key observation is that non-i.i.d. datasets slow down the convergence under the same learning rate settings for both cases. The higher the non-i.i.d. degree, the slower the convergence speed. As the non-i.i.d. degree increases (from case $digits\_10$ to case $digits\_1$), it is obvious that the training loss is increasing and test accuracy is decreasing. This trend is more obvious from the zigzagging curves for partial worker participation. These two observations can also be verified for other models as shown in the supplementary material, which confirms our theoretical analysis.

**Impact of worker number:** As shown in Figure 1(b), we compare the training loss and test accuracy between full worker participation $n = 100$ and partial worker participation $n = 10$ with the same hyper-parameters. Compared with full worker participation, partial worker participation introduces another source of randomness, which leads to zigzagging convergence curves and slower convergence. This problem is more prominent for highly non-i.i.d. datasets. For full worker participation, it can neutralize the the system heterogeneity in each communication round. However, it might not be able to neutralize the gaps among different workers for partial worker participation. That is, the datasets' distribution does not approximate the overall distribution well. Specifically, it is not unlikely that the digits in these datasets among all active workers are only a proper subset of the total 10 digits in the original MNIST dataset, especially with highly non-i.i.d. datasets. This trend is also obvious for complex models and complicated datasets as shown in the supplementary material. The sampling strategy here is random sampling with equal probability without replacement. In practice, however, the actual sampling of the workers in FL could be more complex, which requires further investigations.

**Impact of local steps:** One open question of FL is that whether the local steps help the convergence or not. In Figure 1(c), we show that the local steps could help the convergence for both full and partial worker participation. These results verify our theoretical analysis. However, Li et al. (2019b) showed that the local steps may hurt the convergence, which was demonstrated under unbalanced non-i.i.d. MNIST datasets. We believe that this may be due to the combined effect of unbalanced datasets and local steps rather than just the use of local steps only.

Table 2: Comparison with SCAFFOLD.

| Dataset | IID or Non-IID | Worker selected | Model | SCAFFOLD | | | This paper | | |
|---|---|---|---|---|---|---|---|---|---|
| | | | | # of Round | Communication cost (MB) | Wall-clock time (s) | # of Round | Communication cost (MB) | Wall-clock time (s) |
| MNIST | IID | $n = 10$ | Logistic | 3 | 0.36 | 0.32 | 3 | 0.18 | 0.22 |
| | | | 2NN | 3 | 9.12 | 0.88 | 3 | 4.56 | 0.56 |
| | | | CNN | 3 | 26.64 | 2.23 | 3 | 13.32 | 1.57 |
| | | $n = 100$ | Logistic | 5 | 0.60 | 0.53 | 5 | 0.30 | 0.42 |
| | | | 2NN | 5 | 15.20 | 1.51 | 8 | 12.16 | 1.49 |
| | | | CNN | 1 | 8.88 | 0.79 | 1 | 4.44 | 0.50 |
| | Non-IID | $n = 10$ | Logistic | 14 | 1.68 | 1.48 | 14 | 0.84 | 1.16 |
| | | | 2NN | 14 | 42.55 | 4.23 | 14 | 21.28 | 2.46 |
| | | | CNN | 14 | 124.34 | 11.12 | 10 | 44.41 | 4.92 |
| | | $n = 100$ | Logistic | 7 | 0.84 | 0.72 | 11 | 0.66 | 0.91 |
| | | | 2NN | 7 | 21.28 | 2.11 | 17 | 25.84 | 3.16 |
| | | | CNN | 17 | 150.98 | 13.50 | 7 | 31.08 | 3.51 |
| CIFAR-10 | IID | $n = 10$ | Resnet18 | 56 | 9548.07 | 583.24 | 44 | 3751.03 | 256.63 |
| | Non-IID | $n = 10$ | Resnet18 | 52 | 8866.06 | 539.50 | 61 | 5200.29 | 358.22 |

Bandwidth = 20MB/s.

**Comparison with SCAFFOLD:** Lastly, we compare with the SCAFFOLD algorithm (Karimireddy et al., 2019) since it also achieves the same linear speedup effect under non-i.i.d. datasets. We compare communication rounds, total communication load, and estimated wall-clock time under the same settings to achieve certain test accuracy, and the results are reported in Table 2. The non-i.i.d. dataset is $digits\_2$ and the i.i.d. dataset is $digits\_10$. The learning rates are $\eta_L = 0.1, \eta = 1.0$, and number of local steps $K$ is 5 epochs. We set the target accuracy $\epsilon = 95\%$ for MNIST and $\epsilon = 75\%$ for CIFAR-10. Note that the total training time contains two parts: i) the computation time for training the local model at each worker and ii) the communication time for information exchanges between the workers and the server. We assume the bandwidth 20 MB/s for both uplink and downlink connections. For MNIST datasets, we can see that our algorithm is similar to or outperforms SCAFFOLD. This is because the numbers of communication rounds of both algorithms are relatively small for such simple tasks. For non-i.i.d. CIFAR-10, the SCAFFOLD algorithm takes slightly fewer number of communication rounds than our FedAvg algorithm to achieve $\epsilon = 75\%$ thanks to its variance reduction. However, it takes more than *1.5 times* of communication cost and wall-clock time compared to those of our FedAvg algorithm. Due to space limitation, we relegate the results of time proportions for computation and communication to Appendix B (see Figure 7).

# 6 CONCLUSIONS AND FUTURE WORK

In this paper, we analyzed the convergence of a generlized FedAvg algorithm with two-sided learning rates on non-i.i.d. datasets for general non-convex optimization. We proved that the generalized FedAvg algorithm achieves a linear speedup for convergence under full and partial worker participation. We showed that the local steps in FL could help the convergence and we improve the maximum number of local steps to $T/m$. While our work sheds light on theoretical understanding of FL, it also opens the doors to many new interesting questions in FL, such as how to sample optimally in partial worker participation, and how to deal with active participant sets that are both time-varying and size-varying across communication rounds. We hope that the insights and proof techniques in this paper can pave the way for many new research directions in the aforementioned areas.

## ACKNOWLEDGEMENTS

This work is supported in part by NSF grants CAREER CNS-1943226, CIF-2110252, ECCS-1818791, CCF-1934884, ONR grant ONR N00014-17-1-2417, and a Google Faculty Research Award.

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

# A APPENDIX I: PROOFS

In this section, we give the proofs in detail for full and partial worker participation in Section A.1 and Section A.2, respectively.

## A.1 PROOF OF THEOREM 1

**Theorem 1.** *Let constant local and global learning rates $\eta_L$ and $\eta$ be chosen as such that $\eta_L \leq \frac{1}{8LK}$ and $\eta\eta_L \leq \frac{1}{KL}$. Under Assumptions 1–3 and with full worker participation, the sequence of outputs $\{\mathbf{x}_k\}$ generated by Algorithm 1 satisfies:*

$$\min_{t \in [T]} \mathbb{E}[\|\nabla f(\mathbf{x}_t)\|_2^2] \leq \frac{f_0 - f_*}{c\eta\eta_L KT} + \Phi,$$

*where $\Phi \triangleq \frac{1}{c}[\frac{L\eta\eta_L}{2m}\sigma_L^2 + \frac{5K\eta_L^2 L^2}{2}(\sigma_L^2 + 6K\sigma_G^2)]$, $c$ is a constant, $f_0 \triangleq f(\mathbf{x}_0)$, $f_* \triangleq f(\mathbf{x}_*)$ and the expectation is over the local dataset samples among workers.*

*Proof.* For convenience, we define $\bar{\Delta}_t \triangleq \frac{1}{m}\sum_{i=1}^{m} \Delta_t^i$. Under full device participation (i.e., $S_t = [m]$), it is clear that $\Delta_t = \frac{1}{m}\sum_{i=1}^{m} \Delta_t^i = \bar{\Delta}_t$.

Due to the smoothness in Assumption 1, taking expectation of $f(\mathbf{x}_{t+1})$ over the randomness at communication round $t$, we have:

$$\mathbb{E}_t[f(\mathbf{x}_{t+1})] \leq f(\mathbf{x}_t) + \langle \nabla f(\mathbf{x}_t), \mathbb{E}_t[\mathbf{x}_{t+1} - \mathbf{x}_t]\rangle + \frac{L}{2}\mathbb{E}_t[\|\mathbf{x}_{t+1} - \mathbf{x}_t\|^2]$$

$$= f(\mathbf{x}_t) + \langle \nabla f(\mathbf{x}_t), \mathbb{E}_t[\eta\bar{\Delta}_t + \eta\eta_L K\nabla f(\mathbf{x}_t) - \eta\eta_L K\nabla f(\mathbf{x}_t)]\rangle + \frac{L}{2}\eta^2\mathbb{E}_t[\|\bar{\Delta}_t\|^2]$$

$$= f(\mathbf{x}_t) - \eta\eta_L K\|\nabla f(\mathbf{x}_t)\|^2 + \eta\underbrace{\langle \nabla f(\mathbf{x}_t), \mathbb{E}_t[\bar{\Delta}_t + \eta_L K\nabla f(\mathbf{x}_t)]\rangle}_{A_1} + \frac{L}{2}\eta^2\underbrace{\mathbb{E}_t[\|\bar{\Delta}_t\|^2]}_{A_2}.$$

$$\tag{1}$$

Note that the term $A_1$ in (1) can be bounded as follows:

$$A_1 = \langle \nabla f(\mathbf{x}_t), \mathbb{E}_t[\bar{\Delta}_t + \eta_L K\nabla f(\mathbf{x}_t)]\rangle$$

$$= \left\langle \nabla f(\mathbf{x}_t), \mathbb{E}_t\left[-\frac{1}{m}\sum_{i=1}^{m}\sum_{k=0}^{K-1}\eta_L \mathbf{g}_{t,k}^i + \eta_L K\nabla f(x_t)\right]\right\rangle$$

$$= \left\langle \nabla f(\mathbf{x}_t), \mathbb{E}_t\left[-\frac{1}{m}\sum_{i=1}^{m}\sum_{k=0}^{K-1}\eta_L \nabla F_i(\mathbf{x}_{t,k}^i) + \eta_L K\frac{1}{m}\sum_{i=1}^{m}\nabla F_i(\mathbf{x}_t)\right]\right\rangle$$

$$= \left\langle \sqrt{\eta_L K}\nabla f(\mathbf{x}_t), -\frac{\sqrt{\eta_L}}{m\sqrt{K}}\mathbb{E}_t\sum_{i=1}^{m}\sum_{k=0}^{K-1}(\nabla F_i(\mathbf{x}_{t,k}^i) - \nabla F_i(\mathbf{x}_t))\right\rangle$$

$$\overset{(a1)}{=} \frac{\eta_L K}{2}\|\nabla f(\mathbf{x}_t)\|^2 + \frac{\eta_L}{2Km^2}\mathbb{E}_t\left\|\sum_{i=1}^{m}\sum_{k=0}^{K-1}\nabla F_i(\mathbf{x}_{t,k}^i) - \nabla F_i(\mathbf{x}_t)\right\|^2 - \frac{\eta_L}{2Km^2}\mathbb{E}_t\left\|\sum_{i=1}^{m}\sum_{k=0}^{K-1}\nabla F_i(\mathbf{x}_{t,k}^i)\right\|^2$$

$$\overset{(a2)}{\leq} \frac{\eta_L K}{2}\|\nabla f(\mathbf{x}_t)\|^2 + \frac{\eta_L}{2m}\sum_{i=1}^{m}\sum_{k=0}^{K-1}\mathbb{E}_t\|\nabla F_i(\mathbf{x}_{t,k}^i) - \nabla F_i(\mathbf{x}_t)\|^2 - \frac{\eta_L}{2Km^2}\mathbb{E}_t\left\|\sum_{i=1}^{m}\sum_{k=0}^{K-1}\nabla F_i(\mathbf{x}_{t,k}^i)\right\|^2$$

$$\overset{(a3)}{\leq} \frac{\eta_L K}{2}\|\nabla f(\mathbf{x}_t)\|^2 + \frac{\eta_L L^2}{2m}\sum_{i=1}^{m}\sum_{k=0}^{K-1}\mathbb{E}_t\|\mathbf{x}_{t,k}^i - \mathbf{x}_t\|^2 - \frac{\eta_L}{2Km^2}\mathbb{E}_t\left\|\sum_{i=1}^{m}\sum_{k=0}^{K-1}\nabla F_i(\mathbf{x}_{t,k}^i)\right\|^2$$

$$\overset{(a4)}{\leq} \eta_L K(\frac{1}{2} + 15K^2\eta_L^2 L^2)\|\nabla f(x_t)\|^2 + \frac{5K^2\eta_L^3 L^2}{2}(\sigma_L^2 + 6K\sigma_G^2) - \frac{\eta_L}{2Km^2}\mathbb{E}_t\left\|\sum_{i=1}^{m}\sum_{k=0}^{K-1}\nabla F_i(\mathbf{x}_{t,k}^i)\right\|^2,$$

$$\tag{2}$$

where $(a1)$ follows from that $\langle \mathbf{x}, \mathbf{y} \rangle = \frac{1}{2}[\|\mathbf{x}\|^2 + \|\mathbf{y}\|^2 - \|\mathbf{x} - \mathbf{y}\|^2]$ for $\mathbf{x} = \sqrt{\eta_L K} \nabla f(\mathbf{x}_t)$ and $\mathbf{y} = -\frac{\sqrt{\eta_L}}{m\sqrt{K}} \sum_{i=1}^{m} \sum_{k=0}^{K-1} (\nabla F_i(\mathbf{x}_{t,k}^i) - \nabla F_i(\mathbf{x}_t))$, $(a2)$ is due to that $\mathbb{E}[\|x_1 + \cdots + x_n\|^2] \leq n\mathbb{E}[\|x_1\|^2 + \cdots + \|x_n\|^2]$, $(a3)$ is due to Assumption 1 and $(a4)$ follows from Lemma 2.

The term $A_2$ in (1) can be bounded as:

$$
\begin{aligned}
A_2 &= \mathbb{E}_t[\|\bar{\Delta}_t\|^2] \\
&= \mathbb{E}_t\left[\left\|\frac{1}{m}\sum_{i=1}^{m}\Delta_t^i\right\|^2\right] \\
&\leq \frac{1}{m^2}\mathbb{E}_t\left[\left\|\sum_{i=1}^{m}\Delta_t^i\right\|^2\right] \\
&= \frac{\eta_L^2}{m^2}\mathbb{E}_t\left[\left\|\sum_{i=1}^{m}\sum_{k=0}^{K-1}\mathbf{g}_{t,k}^i\right\|^2\right] \\
&\overset{(a5)}{=} \frac{\eta_L^2}{m^2}\mathbb{E}_t\left[\left\|\sum_{i=1}^{m}\sum_{k=0}^{K-1}(\mathbf{g}_{t,k}^i - \nabla F_i(\mathbf{x}_{t,k}^i))\right\|^2\right] + \frac{\eta_L^2}{m^2}\mathbb{E}_t\left\|\sum_{i=1}^{m}\sum_{k=0}^{K-1}\nabla F_i(\mathbf{x}_{t,k}^i)\right\|^2 \\
&\overset{(a6)}{\leq} \frac{K\eta_L^2}{m}\sigma_L^2 + \frac{\eta_L^2}{m^2}\mathbb{E}_t\left\|\sum_{i=1}^{m}\sum_{k=0}^{K-1}\nabla F_i(\mathbf{x}_{t,k}^i)\right\|^2,
\end{aligned}
\tag{3}
$$

where $(a5)$ follows from the fact that $\mathbb{E}[\|\mathbf{x}\|^2] = \mathbb{E}[\|\mathbf{x} - \mathbb{E}[\mathbf{x}]\|^2] + \|\mathbb{E}[\mathbf{x}]\|^2$ and $(a6)$ is due to the bounded variance assumption in Assumption 3 and the fact that $\mathbb{E}[\|x_1 + \cdots + x_n\|^2] = \mathbb{E}[\|x_1\|^2 + \cdots + \|x_n\|^2]$ if $x_i'$s are independent with zero mean and $\mathbb{E}[\mathbf{g}_{t,j}^i] = \nabla F_i(\mathbf{x}_{t,j}^i)$.

Substituting the inequalities in (2) of $A_1$ and (3) of $A_2$ into inequality (1), we have:

$$
\begin{aligned}
\mathbb{E}_t[f(\mathbf{x}_{t+1})] &\leq f(\mathbf{x}_t) - \eta\eta_L K\|\nabla f(\mathbf{x}_t)\|^2 + \eta\underbrace{< \nabla f(\mathbf{x}_t), \mathbb{E}_t[\bar{\Delta}_t + \eta_L K \nabla f(\mathbf{x}_t)] >}_{A_1} + \frac{L}{2}\eta^2\underbrace{\mathbb{E}_t[\|\bar{\Delta}_t\|^2]}_{A_2} \\
&\leq f(\mathbf{x}_t) - \eta\eta_L K(\frac{1}{2} - 15K^2\eta_L^2 L^2)\|\nabla f(\mathbf{x}_t)\|^2 + \frac{LK\eta^2\eta_L^2}{2m}\sigma_L^2 \\
&\quad + \frac{5\eta K^2\eta_L^3 L^2}{2}(\sigma_L^2 + 6K\sigma_G^2) - (\frac{\eta\eta_L}{2Km^2} - \frac{L\eta^2\eta_L^2}{2m^2})\mathbb{E}_t\left\|\sum_{i=1}^{m}\sum_{k=0}^{K-1}\nabla F_i(\mathbf{x}_{t,k}^i)\right\|^2 \\
&\overset{(a7)}{\leq} f(\mathbf{x}_t) - \eta\eta_L K(\frac{1}{2} - 5K^2\eta_L^2 L^2)\|\nabla f(\mathbf{x}_t)\|^2 + \frac{LK\eta^2\eta_L^2}{2m}\sigma_L^2 + \frac{5\eta K^2\eta_L^3 L^2}{2}(\sigma_L^2 + 6K\sigma_G^2) \\
&\overset{(a8)}{\leq} f(\mathbf{x}_t) - c\eta\eta_L K\|\nabla f(\mathbf{x}_t)\|^2 + \frac{LK\eta^2\eta_L^2}{2m}\sigma_L^2 + \frac{5\eta K^2\eta_L^3 L^2}{2}(\sigma_L^2 + 6K\sigma_G^2),
\end{aligned}
$$

where $(a7)$ follows from $(\frac{\eta\eta_L}{2Km^2} - \frac{L\eta^2\eta_L^2}{2m^2}) \geq 0$ if $\eta\eta_L \leq \frac{1}{KL}$, $(a8)$ holds because there exists a constant $c > 0$ satisfying $(\frac{1}{2} - 15K^2\eta_L^2 L^2) > c > 0$ if $\eta_L < \frac{1}{\sqrt{30}KL}$.

Rearranging and summing from $t = 0, \cdots, T - 1$, we have:

$$
\sum_{t=0}^{T-1} c\eta\eta_L K\mathbb{E}[\nabla f(\mathbf{x}_t)] \leq f(\mathbf{x}_0) - f(\mathbf{x}_T) + T(\eta\eta_L K)\left[\frac{L\eta\eta_L}{2m}\sigma_L^2 + \frac{5K\eta_L^2 L^2}{2}(\sigma_L^2 + 6K\sigma_G^2)\right]
$$

which implies,

$$
\min_{t\in[T]}\mathbb{E}\|\nabla f(\mathbf{x}_t)\|_2^2 \leq \frac{f_0 - f_*}{c\eta\eta_L KT} + \Phi,
$$

where $\Phi = \frac{1}{c}\left[\frac{L\eta\eta_L}{2m}\sigma_L^2 + \frac{5K\eta_L^2 L^2}{2}(\sigma_L^2 + 6K\sigma_G^2)\right]$. This completes the proof. $\qquad\square$

## A.2   PROOF OF THEOREM 2

**Theorem 2.** *Under Assumptions 1–3 with partial worker participation, the sequence of outputs $\{\mathbf{x}_k\}$ generated by Algorithm 1 with constant learning rates $\eta$ and $\eta_L$ satisfies:*

$$\min_{t \in [T]} \mathbb{E}[\|\nabla f(\mathbf{x}_t)\|_2^2] \leq \frac{f_0 - f_*}{c \eta \eta_L K T} + \Phi,$$

*where $f_0 = f(\mathbf{x}_0)$, $f_* = f(\mathbf{x}_*)$, and the expectation is over the local dataset samples among workers.*

*For sampling Strategy 1, let $\eta$ and $\eta_L$ be chosen as such that $\eta_L \leq \frac{1}{8LK}$, $\eta \eta_L K L < \frac{n-1}{n}$ and $30 K^2 \eta_L^2 L^2 - \frac{L \eta \eta_L}{n}(90 K^3 L^2 \eta_L^2 + 3K) < 1$. It then holds that:*

$$\Phi \triangleq \frac{1}{c}\left[ \frac{L \eta \eta_L}{2n} \sigma_L^2 + \frac{3 L K \eta \eta_L}{2n} \sigma_G^2 + \left( \frac{5 K \eta_L^2 L^2}{2} + \frac{15 K^2 \eta \eta_L^3 L^3}{2n} \right)(\sigma_L^2 + 6 K \sigma_G^2) \right].$$

*For sampling Strategy 2, let $\eta$ and $\eta_L$ be chosen as such that $\eta_L \leq \frac{1}{8LK}$, $\eta \eta_L K L \leq \frac{n(m-1)}{m(n-1)}$ and $10 K^2 \eta_L^2 L^2 - L \eta \eta_L \frac{m-n}{n(m-1)}(90 K^3 \eta_L^2 L^2 + 3K) < 1$. It then holds that:*

$$\Phi \triangleq \frac{1}{c}\left[ \frac{L \eta \eta_L}{2n} \sigma_L^2 + 3 L K \eta \eta_L \frac{m-n}{2n(m-1)} \sigma_G^2 + \left( \frac{5 K \eta_L^2 L^2}{2} + 15 K^2 \eta \eta_L^3 L^3 \frac{m-n}{2n(m-1)} \right)(\sigma_L^2 + 6 K \sigma_G^2) \right].$$

*Proof.* Let $\bar{\Delta}_t$ be defined the same as in the proof of Theorem 1. Under partial device participation, note that $\bar{\Delta}_t \neq \Delta_t$ (recall that $\bar{\Delta}_t \triangleq \frac{1}{m} \sum_{i=1}^m \Delta_t^i$, $\Delta_t = \frac{1}{n} \sum_{i \in S_t} \Delta_t^i$, and $|S_t| = n$). The randomness for partial worker participation contains two parts: the random sampling and the stochastic gradient. We still use $\mathbb{E}_t[\cdot]$ to represent the expectation with respect to both types of randomness.

Due to the smoothness assumption in Assumption 1, taking expectation of $f(\mathbf{x}_{t+1})$ over the randomness at communication round t:

$$\mathbb{E}_t[f(\mathbf{x}_{t+1})] \leq f(\mathbf{x}_t) + \langle \nabla f(\mathbf{x}_t), \mathbb{E}_t[\mathbf{x}_{t+1} - \mathbf{x}_t] \rangle + \frac{L}{2} \mathbb{E}_t[\|\mathbf{x}_{t+1} - \mathbf{x}_t\|^2]$$

$$= f(\mathbf{x}_t) + \langle \nabla f(\mathbf{x}_t), \mathbb{E}_t[\eta \Delta_t + \eta \eta_L K \nabla f(\mathbf{x}_t) - \eta \eta_L K \nabla f(\mathbf{x}_t)] \rangle + \frac{L}{2} \eta^2 \mathbb{E}_t[\|\Delta_t\|^2]$$

$$= f(\mathbf{x}_t) - \eta \eta_L K \|\nabla f(\mathbf{x}_t)\|^2 + \eta \underbrace{\langle \nabla f(\mathbf{x}_t), \mathbb{E}_t[\Delta_t + \eta_L K \nabla f(\mathbf{x}_t)] \rangle}_{A_1'} + \frac{L}{2} \eta^2 \underbrace{\mathbb{E}_t[\|\Delta_t\|^2]}_{A_2'}$$

$$\tag{4}$$

The term $A_1'$ in (4) can be bounded as follows: Since $\mathbb{E}_{S_t}[A_1'] = A_1$ due to Lemma 1 for both sampling strategies, we have the same bound as in inequality 2 for $A_1'$:

$$A_1' \leq \eta_L K \left( \frac{1}{2} + 15 K^2 \eta_L^2 L^2 \right) \|\nabla f(x_t)\|^2 + \frac{5 K^2 \eta_L^3 L^2}{2}(\sigma_L^2 + 6 K \sigma_G^2)$$

$$- \frac{\eta_L}{2 K m^2} \mathbb{E}_t \left\| \sum_{i=1}^m \sum_{k=0}^{K-1} \nabla F_i(\mathbf{x}_{t,k}^i) \right\|^2, \quad (5)$$

**For strategy 1:** We can bound $A_2'$ in (4) as follows.

Note $S_t$ is an index set (multiset) for independent sampling (equal probability) with replacement in which some elements may have the same value. Suppose $S_t = \{l_1, \ldots, l_n\}$.

$$A_2' = \mathbb{E}_t[\|\Delta_t\|^2]$$

$$= \mathbb{E}_t \left[ \left\| \frac{1}{n} \sum_{i \in S_t} \Delta_t^i \right\|^2 \right]$$

$$= \frac{1}{n^2} \mathbb{E}_t \left[ \left\| \sum_{i \in S_t} \Delta_t^i \right\|^2 \right]$$

$$= \frac{1}{n^2} \mathbb{E}_t \left[ \left\| \sum_{z=1}^n \Delta_t^{l_z} \right\|^2 \right]$$

$$\overset{(b1)}{=} \frac{\eta_L^2}{n^2} \mathbb{E}_t \left[ \left\| \sum_{z=1}^n \sum_{j=0}^{K-1} [\mathbf{g}_{t,j}^{l_z} - \nabla F_{l_z}(\mathbf{x}_{t,j}^{l_z})] \right\|^2 \right] + \frac{\eta_L^2}{n^2} \mathbb{E}_t \left[ \left\| \sum_{z=1}^n \sum_{j=0}^{K-1} \nabla F_{l_z}(\mathbf{x}_{t,j}^{l_z}) \right\|^2 \right]$$

$$\overset{(b2)}{\leq} \frac{K\eta_L^2}{n} \sigma_L^2 + \frac{\eta_L^2}{n^2} \mathbb{E}_t \left[ \left\| \sum_{z=1}^n \sum_{j=0}^{K-1} \nabla F_{l_z}(\mathbf{x}_{t,j}^{l_z}) \right\|^2 \right],$$

where $(b1)$ follows from the fact that $\mathbb{E}[\|\mathbf{x}\|^2] = \mathbb{E}[\|\mathbf{x} - \mathbb{E}[\mathbf{x}]\|^2] + \|\mathbb{E}[\mathbf{x}]\|^2$ and $(b2)$ is due to the bounded variance assumption 3 and $\mathbb{E}[\|x_1 + \cdots + x_n\|^2] \leq n\mathbb{E}[\|x_1\|^2 + \cdots + \|x_n\|^2]$.

By letting $\mathbf{t}_i = \sum_{j=0}^{K-1} \nabla F_i(\mathbf{x}_{t,j}^i)$, we have:

$$\mathbb{E}_t \left[ \left\| \sum_{z=1}^n \sum_{j=0}^{K-1} \nabla F_{l_z}(\mathbf{x}_{t,j}^{l_z}) \right\|^2 \right] = \mathbb{E}_t \left[ \left\| \sum_{z=1}^n \mathbf{t}_{l_z} \right\|^2 \right]$$

$$= \mathbb{E}_t \left[ \sum_{z=1}^n \|\mathbf{t}_{l_z}\|^2 + \sum_{i \neq j; l_i, l_j \in S_t} \langle \mathbf{t}_{l_i}, \mathbf{t}_{l_j} \rangle \right]$$

$$\overset{(b3)}{=} \mathbb{E}_t \left[ n\|\mathbf{t}_{l_1}\|^2 + n(n-1)\langle \mathbf{t}_{l_1}, \mathbf{t}_{l_2} \rangle \right]$$

$$= \frac{n}{m} \sum_{i=1}^m \|\mathbf{t}_i\|^2 + \frac{n(n-1)}{m^2} \sum_{i,j \in [m]} \langle \mathbf{t}_i, \mathbf{t}_j \rangle$$

$$= \frac{n}{m} \sum_{i=1}^m \|\mathbf{t}_i\|^2 + \frac{n(n-1)}{m^2} \| \sum_{i=1}^m \mathbf{t}_i \|^2,$$

where $(b3)$ is due to the independent sampling with replacement.

So we can bound $A_2'$ as follows.

$$A_2' = \mathbb{E}_t[\|\Delta_t\|^2]$$

$$\leq \frac{K\eta_L^2}{n}\sigma_L^2 + \frac{\eta_L^2}{mn} \sum_{i=1}^m \mathbb{E}_t\|\mathbf{t}_i\|^2 + \frac{(n-1)\eta_L^2}{m^2 n} \mathbb{E}_t \left\| \sum_{i=1}^m \mathbf{t}_i \right\|^2, \tag{6}$$

For $\mathbf{t}_i$, we have:

$$\sum_{i=1}^m \mathbb{E}_t\|\mathbf{t}_i\|^2 = \sum_{i=1}^m \mathbb{E}_t \left\| \sum_{j=0}^{K-1} \nabla F_i(\mathbf{x}_{t,j}^i) - \nabla F_i(\mathbf{x}_t) + \nabla F_i(\mathbf{x}_t) - \nabla f(\mathbf{x}_t) + \nabla f(\mathbf{x}_t) \right\|^2$$

$$\overset{(b4)}{\leq} 3KL^2 \sum_{i=1}^m \sum_{j=0}^{K-1} \mathbb{E}_t\|\mathbf{x}_{t,j}^i - \mathbf{x}_t\|^2 + 3mK^2\sigma_G^2 + 3mK^2\|\nabla f(\mathbf{x}_t)\|^2$$

$$\overset{(b5)}{\leq} 15mK^3L^2\eta_L^2(\sigma_L^2 + 6K\sigma_G^2) + (90mK^4L^2\eta_L^2 + 3mK^2)\|\nabla f(\mathbf{x}_t)\|^2 + 3mK^2\sigma_G^2, \tag{7}$$

where $(b4)$ is due to the fact that $\mathbb{E}[\|x_1 + \cdots + x_n\|^2] \leq n\mathbb{E}[\|x_1\|^2 + \cdots + \|x_n\|^2]$, Assumptions 3 and 1, and $(b5)$ follows from Lemma 2.

Substituting the inequalities in ( 5) of $A_1'$ and ( 6) of $A_2'$ into inequality (4), we have:

$$\mathbb{E}_t[f(\mathbf{x}_{t+1})] \leq f(\mathbf{x}_t) - \eta\eta_L K \|\nabla f(\mathbf{x}_t)\|^2 + \eta \underbrace{\langle \nabla f(\mathbf{x}_t), \mathbb{E}_t[\Delta_t + \eta_L K \nabla f(\mathbf{x}_t)]\rangle}_{A_1'} + \frac{L}{2}\eta^2 \underbrace{\mathbb{E}_t[\|\Delta_t\|^2]}_{A_2'}$$

$$\leq f(\mathbf{x}_t) - \eta\eta_L K(\frac{1}{2} - 15K^2\eta_L^2 L^2)\|\nabla f(x_t)\|^2 + \frac{5\eta K^2 \eta_L^3 L^2}{2}(\sigma_L^2 + 6K\sigma_G^2)$$

$$+ \left[\frac{(n-1)L\eta^2\eta_L^2}{2m^2 n} - \frac{\eta\eta_L}{2Km^2}\right]\mathbb{E}_t\left\|\sum_{i=1}^m \mathbf{t}_i\right\|^2 + \frac{LK\eta^2\eta_L^2}{2n}\sigma_L^2 + \frac{L\eta^2\eta_L^2}{2mn}\sum_{i=1}^m \mathbb{E}_t\|\mathbf{t}_i\|^2$$

$$\overset{(b6)}{\leq} f(\mathbf{x}_t) - \eta\eta_L K(\frac{1}{2} - 15K^2\eta_L^2 L^2)\|\nabla f(x_t)\|^2 + \frac{5\eta K^2 \eta_L^3 L^2}{2}(\sigma_L^2 + 6K\sigma_G^2)$$

$$+ \frac{LK\eta^2\eta_L^2}{2n}\sigma_L^2 + \frac{L\eta^2\eta_L^2}{2mn}\sum_{i=1}^m \mathbb{E}_t\|\mathbf{t}_i\|^2$$

$$\overset{(b7)}{\leq} f(\mathbf{x}_t) - \eta\eta_L K(\frac{1}{2} - 15K^2\eta_L^2 L^2 - \frac{L\eta\eta_L}{2n}(90K^3 L^2 \eta_L^2 + 3K))\|\nabla f(x_t)\|^2$$

$$+ \left[\frac{5\eta K^2 \eta_L^3 L^2}{2} + \frac{15K^3 L^3 \eta^2 \eta_L^4}{2n}\right](\sigma_L^2 + 6K\sigma_G^2) + \frac{LK\eta^2\eta_L^2}{2n}\sigma_L^2 + \frac{3K^2 L\eta^2\eta_L^2}{2n}\sigma_G^2$$

$$\overset{(b8)}{\leq} f(\mathbf{x}_t) - c\eta\eta_L K \|\nabla f(\mathbf{x}_t)\|^2 + \frac{LK\eta^2\eta_L^2}{2n}\sigma_L^2 + \frac{3K^2 L\eta^2\eta_L^2}{2n}\sigma_G^2$$

$$+ \eta\eta_L K \left[\frac{5K\eta_L^2 L^2}{2} + \frac{15K^2 \eta_L^3 \eta L^3}{2n}\right](\sigma_L^2 + 6K\sigma_G^2), \tag{8}$$

where ($b6$) follows from $\frac{(n-1)L\eta^2\eta_L^2}{2m^2 n} - \frac{\eta\eta_L}{2Km^2} \leq 0$ if $\eta\eta_L KL \leq \frac{n-1}{n}$, ($b7$)is due to inequality (7) and ($b8$) holds since there exists a constant $c > 0$ such that $[\frac{1}{2} - 15K^2\eta_L^2 L^2 - \frac{L\eta\eta_L}{2n}(90K^3 L^2 \eta_L^2 + 3K)] > c > 0$ if $30K^2\eta_L^2 L^2 - \frac{L\eta\eta_L}{n}(90K^3 L^2 \eta_L^2 + 3K) < 1$.

Note that the requirement of $|S_t| = n$ can be relaxed to $|S_t| \geq n$. With $p_t \geq n$ workers in $t$-th communication round, 8 is

$$\mathbb{E}_t[f(\mathbf{x}_{t+1})] \leq f(\mathbf{x}_t) - c\eta\eta_L K \|\nabla f(\mathbf{x}_t)\|^2 + \frac{LK\eta^2\eta_L^2}{2p_t}\sigma_L^2 + \frac{3KL\eta^2\eta_L^2}{2p_t}\sigma_G^2$$

$$+ \eta\eta_L K \left[\frac{5K\eta_L^2 L^2}{2} + \frac{15K\eta_L^3 \eta L^3}{2p_t}\right](\sigma_L^2 + 6K\sigma_G^2)$$

$$\leq f(\mathbf{x}_t) - c\eta\eta_L K \|\nabla f(\mathbf{x}_t)\|^2 + \frac{LK\eta^2\eta_L^2}{2n}\sigma_L^2 + \frac{3K^2 L\eta^2\eta_L^2}{2n}\sigma_G^2$$

$$+ \eta\eta_L K \left[\frac{5K\eta_L^2 L^2}{2} + \frac{15K^2 \eta_L^3 \eta L^3}{2n}\right](\sigma_L^2 + 6K\sigma_G^2).$$

That is, the same convergence rate can be guaranteed if at least $n$ workers in each communication round (no need to be exactly $n$).

Rearranging and summing from $t = 0, \cdots, T - 1$, we have the convergence for partial device participation with sampling strategy 1 as follows:

$$\min_{t\in[T]} \mathbb{E}[\|\nabla f(\mathbf{x}_t)\|_2^2] \leq \frac{f_0 - f_*}{c\eta\eta_L KT} + \Phi,$$

where $\Phi = \frac{1}{c}\left[\frac{L\eta\eta_L}{2n}\sigma_L^2 + \frac{3KL\eta\eta_L}{2n}\sigma_G^2 + (\frac{5K\eta_L^2 L^2}{2} + \frac{15K^2 \eta\eta_L^3 L^3}{2n})(\sigma_L^2 + 6K\sigma_G^2)\right]$ and $c$ is a constant.

**For strategy 2:** Under the strategy of independent sampling with equal probability without replacement. We bound $A_2'$ as follows.

$$A_2' = \mathbb{E}_t[\|\Delta_t\|^2]$$

$$= \mathbb{E}_t\left[\left\|\frac{1}{n}\sum_{i\in S_t} \Delta_t^i\right\|^2\right]$$

$$= \frac{1}{n^2} \mathbb{E}_t \left[ \left\| \sum_{i \in S_t} \Delta_t^i \right\|^2 \right]$$

$$= \frac{1}{n^2} \mathbb{E}_t \left[ \left\| \sum_{i=1}^{m} \mathbb{I}\{i \in S_t\} \Delta_t^i \right\|^2 \right]$$

$$= \frac{\eta_L^2}{n^2} \mathbb{E}_t \left[ \left\| \sum_{i=1}^{m} \mathbb{I}\{i \in S_t\} \sum_{j=0}^{K-1} [\mathbf{g}_{t,j}^i - \nabla F_i(\mathbf{x}_{t,j}^i)] \right\|^2 \right] + \frac{\eta_L^2}{n^2} \mathbb{E}_t \left[ \left\| \sum_{i=1}^{m} \mathbb{I}\{i \in S_t\} \sum_{j=0}^{K-1} \nabla F_i(\mathbf{x}_{t,j}^i)] \right\|^2 \right]$$

$$= \frac{\eta_L^2}{n^2} \mathbb{E}_t \left[ \left\| \sum_{i=1}^{m} \mathbb{P}\{i \in S_t\} \sum_{j=0}^{K-1} [\mathbf{g}_{t,j}^i - \nabla F_i(\mathbf{x}_{t,j}^i)] \right\|^2 + \frac{\eta_L^2}{n^2} \left\| \sum_{i=1}^{m} \mathbb{I}\{i \in S_t\} \sum_{j=0}^{K-1} \nabla F_i(\mathbf{x}_{t,j}^i) \right\|^2 \right]$$

$$\overset{(b9)}{=} \frac{\eta_L^2}{nm} \mathbb{E}_t \left[ \sum_{i=1}^{m} \sum_{j=0}^{K-1} \left\| \mathbf{g}_{t,j}^i - \nabla F_i(\mathbf{x}_{t,j}^i) \right\|^2 \right] + \frac{\eta_L^2}{n^2} \mathbb{E}_t \left[ \left\| \sum_{i=1}^{m} \mathbb{I}\{i \in S_t\} \sum_{j=0}^{K-1} \nabla F_i(\mathbf{x}_{t,j}^i) \right\|^2 \right]$$

$$\overset{(b10)}{\leq} \frac{K\eta_L^2}{n} \sigma_L^2 + \frac{\eta_L^2}{n^2} \left\| \sum_{i=1}^{m} \mathbb{P}\{i \in S_t\} \sum_{j=0}^{K-1} \nabla F_i(\mathbf{x}_{t,j}^i) \right\|^2, \tag{9}$$

where $(b9)$ is due to the fact that $\mathbb{E}[\|x_1 + \cdots + x_n\|^2] = \mathbb{E}[\|x_1\|^2 + \cdots + \|x_n\|^2]$ if $x_i'$s are independent with zero mean, $\mathbf{x}_i = \mathbf{g}_{t,j}^i - \nabla F_i(\mathbf{x}_{t,j}^i)$ is independent random variable with mean zero, and $\mathbb{P}\{i \in S_t\} = \frac{n}{m}$. $(b10)$ is due to bounded variance assumption in Assumption 3

Substituting the inequalities in (5) of $A_1'$ and (9) of $A_2'$ into inequality (4), we have:

$$\mathbb{E}_t[f(\mathbf{x}_{t+1})] \leq f(\mathbf{x}_t) - \eta\eta_L K \|\nabla f(\mathbf{x}_t)\|^2 + \eta \underbrace{\langle \nabla f(\mathbf{x}_t), \mathbb{E}_t[\Delta_t + \eta_L K \nabla f(\mathbf{x}_t)] \rangle}_{A_1'} + \frac{L}{2}\eta^2 \underbrace{\mathbb{E}_t[\|\Delta_t\|^2]}_{A_2'}$$

$$\leq \nabla f(\mathbf{x}_t) - \eta\eta_L K(\frac{1}{2} - 15K^2\eta_L^2 L^2) \|\nabla f(\mathbf{x}_t)\|^2 + \frac{LK\eta^2\eta_L^2}{2n}\sigma_L^2 + \frac{5\eta K^2 \eta_L^3 L^2}{2}(\sigma_L^2 + 6K\sigma_G^2)$$

$$+ \underbrace{\frac{L\eta^2\eta_L^2}{2n^2} \mathbb{E}_t \left\| \sum_{i=1}^{m} \mathbb{P}\{i \in S_t\} \sum_{j=0}^{K-1} \nabla F_i(\mathbf{x}_{t,j}^i) \right\|^2 - \frac{\eta\eta_L}{2Km^2} \mathbb{E}_t \left\| \sum_{i=1}^{m} \sum_{k=0}^{K-1} \nabla F_i(\mathbf{x}_{t,k}^i) \right\|^2}_{A_3'}.$$

Then we bound $A_3'$ as follows.

By letting $\mathbf{t}_i = \sum_{j=0}^{K-1} \nabla F_i(\mathbf{x}_{t,j}^i)$, we have:

$$\sum_{i=1}^{m} \mathbb{E}_t \|\mathbf{t}_i\|^2 \leq 15mK^3 L^2 \eta_L^2 (\sigma_L^2 + 6K\sigma_G^2) + (90mK^4 L^2 \eta_L^2 + 3mK^2) \|\nabla f(\mathbf{x}_t)\|^2 + 3mK^2 \sigma_G^2.$$

It then follows that

$$\left\| \sum_{i=1}^{m} \mathbf{t}_i \right\|^2 = \sum_{i \in [m]} \|\mathbf{t}_i\|^2 + \sum_{i \neq j} < \mathbf{t}_i, \mathbf{t}_j >$$

$$\overset{(b11)}{=} \sum_{i \in [m]} m\|\mathbf{t}_i\|^2 - \frac{1}{2} \sum_{i \neq j} \|\mathbf{t}_i - \mathbf{t}_j\|^2$$

$$\left\| \sum_{i=1}^{m} \mathbb{P}\{i \in S_t\} \mathbf{t}_i \right\|^2 = \sum_{i \in [m]} \mathbb{P}\{i \in S_t\} \|\mathbf{t}_i\|^2 + \sum_{i \neq j} \mathbb{P}\{i, j \in S_t\} < \mathbf{t}_i, \mathbf{t}_j >$$

$$\overset{(b12)}{=} \frac{n}{m} \sum_{i \in [m]} \|\mathbf{t}_i\|^2 + \frac{n(n-1)}{m(m-1)} \sum_{i \neq j} < \mathbf{t}_i, \mathbf{t}_j >$$

$$\overset{(b13)}{=} \frac{n^2}{m} \sum_{i \in [m]} \|\mathbf{t}_i\|^2 - \frac{n(n-1)}{2m(m-1)} \sum_{i \neq j} \|\mathbf{t}_i - \mathbf{t}_j\|^2,$$

where $(b11)$ and $(b13)$ are due to the fact that $\langle \mathbf{x}, \mathbf{y} \rangle = \frac{1}{2}[\|\mathbf{x}\|^2 + \|\mathbf{y}\|^2 - \|\mathbf{x} - \mathbf{y}\|^2] \leq \frac{1}{2}[\|\mathbf{x}\|^2 + \|\mathbf{y}\|^2]$, $(b12)$ follows from the fact that $\mathbb{P}\{i \in S_t\} = \frac{n}{m}$ and $\mathbb{P}\{i, j \in S_t\} = \frac{n(n-1)}{m(m-1)}$. Therefore, we have

$$A_3' = \frac{L\eta^2\eta_L^2}{2n^2} \|\sum_{i=1}^{m} \mathbb{P}\{i \in S_t\} \sum_{j=0}^{K-1} \nabla F_i(\mathbf{x}_{t,j}^i)]\|^2 - \frac{\eta\eta_L}{2Km^2} \|\sum_{i=1}^{m} \sum_{k=0}^{K-1} \nabla F_i(\mathbf{x}_{t,k}^i)\|^2$$

$$= (\frac{L\eta^2\eta_L^2}{2m} - \frac{\eta\eta_L}{2Km}) \sum_{i=1}^{m} \|\mathbf{t}_i\|^2 + (\frac{\eta\eta_L}{4Km^2} - \frac{L\eta^2\eta_L^2(n-1)}{4mn(m-1)}) \sum_{i \neq j} \|\mathbf{t}_i - \mathbf{t}_j\|^2$$

$$\overset{(b14)}{=} (\frac{L\eta^2\eta_L^2}{2m} - \frac{L\eta^2\eta_L^2(n-1)}{2n(m-1)}) \sum_{i=1}^{m} \|\mathbf{t}_i\|^2 - (\frac{\eta\eta_L}{2Km^2} - \frac{L\eta^2\eta_L^2(n-1)}{2mn(m-1)}) \|\sum_{i \in [m]} \mathbf{t}_i\|^2$$

$$\overset{(b15)}{\leq} (\frac{L\eta^2\eta_L^2}{2m} - \frac{L\eta^2\eta_L^2(n-1)}{2n(m-1)}) \sum_{i=1}^{m} \|\mathbf{t}_i\|^2$$

$$= L\eta^2\eta_L^2 \frac{m-n}{2mn(m-1)} \sum_{i=1}^{m} \|\mathbf{t}_i\|^2,$$

where $(b14)$ follows from the fact that $\|\sum_{i \in [m]} \mathbf{t}_i\|^2 = \sum_{i \in [m]} m\|\mathbf{t}_i\|^2 - \frac{1}{2} \sum_{i \neq j} \|\mathbf{t}_i - \mathbf{t}_j\|^2$, and $(b15)$ is due to the fact that $(\frac{\eta\eta_L}{2Km^2} - \frac{L\eta^2\eta_L^2(n-1)}{2mn(m-1)}) \geq 0$ if $\eta\eta_L KL \leq \frac{n(m-1)}{m(n-1)}$.

Then we have

$$\mathbb{E}_t[f(\mathbf{x}_{t+1})] \leq f(\mathbf{x}_t) - \eta\eta_L K(\frac{1}{2} - 15K^2\eta_L^2 L^2 - L\eta\eta_L \frac{m-n}{2n(m-1)}(90K^3\eta_L^2 L^2 + 3K))\|\nabla f(\mathbf{x}_t)\|^2$$

$$+ \frac{LK\eta^2\eta_L^2}{2n}\sigma_L^2 + 3K^2 L\eta^2\eta_L^2 \frac{m-n}{2n(m-1)}\sigma_G^2$$

$$+ \eta\eta_L K(\frac{5K\eta_L^2 L^2}{2} + 15K\eta\eta_L^3 L^3 \frac{m-n}{2n(m-1)})(\sigma_L^2 + 6K\sigma_G^2)$$

$$\overset{(b16)}{\leq} f(\mathbf{x}_t) - c\eta\eta_L K\|\nabla f(\mathbf{x}_t)\|^2 + \frac{LK\eta^2\eta_L^2}{2n}\sigma_L^2 + 3KL\eta^2\eta_L^2 \frac{m-n}{2n(m-1)}\sigma_G^2$$

$$+ \eta\eta_L K(\frac{5K\eta_L^2 L^2}{2} + 15K^2\eta\eta_L^3 L^3 \frac{m-n}{2n(m-1)})(\sigma_L^2 + 6K\sigma_G^2), \tag{10}$$

where $(b16)$ holds because there exists a constant $c > 0$ satisfying $(\frac{1}{2} - 5K^2\eta_L^2 L^2 - L\eta\eta_L \frac{m-n}{2n(m-1)}(90K^3\eta_L^2 L^2 + 3K)) > c > 0$ if $10K^2\eta_L^2 L^2 - L\eta\eta_L \frac{m-n}{n(m-1)}(90K^3\eta_L^2 L^2 + 3K) < 1$.

Note that the requirement of $|S_t| = n$ can be relaxed to $|S_t| \geq n$. With $p_t \geq n$ workers in $t$-th communication round, 10 is

$$\mathbb{E}_t[f(\mathbf{x}_{t+1})] \leq f(\mathbf{x}_t) - c\eta\eta_L K\|\nabla f(\mathbf{x}_t)\|^2 + \frac{LK\eta^2\eta_L^2}{2p_t}\sigma_L^2 + 3KL\eta^2\eta_L^2 \frac{m-p_t}{2p_t(m-1)}\sigma_G^2$$

$$+ \eta\eta_L K(\frac{5K\eta_L^2 L^2}{2} + 15K^2\eta\eta_L^3 L^3 \frac{m-p_t}{2p_t(m-1)})(\sigma_L^2 + 6K\sigma_G^2)$$

$$\leq f(\mathbf{x}_t) - c\eta\eta_L K\|\nabla f(\mathbf{x}_t)\|^2 + \frac{LK\eta^2\eta_L^2}{2n}\sigma_L^2 + 3KL\eta^2\eta_L^2 \frac{m-n}{2n(m-1)}\sigma_G^2$$

$$+ \eta\eta_L K(\frac{5K\eta_L^2 L^2}{2} + 15K^2\eta\eta_L^3 L^3 \frac{m-n}{2n(m-1)})(\sigma_L^2 + 6K\sigma_G^2)$$

That is, the same convergence rate can be guaranteed if at least $n$ workers in each communication round (no need to be exactly $n$).

Rearranging and summing from $t = 0, \cdots, T - 1$, we have the convergence for partial device participation with sampling strategy 2 as follows:

$$\min_{t \in [T]} \mathbb{E}[\|\nabla f(\mathbf{x}_t)\|_2^2] \leq \frac{f_0 - f_*}{c\eta\eta_L KT} + \Phi,$$

where $\Phi = \frac{1}{c}\left[\frac{L\eta\eta_L}{2n}\sigma_L^2 + 3KL\eta\eta_L\frac{m-n}{2n(m-1)}\sigma_G^2 + (\frac{5K\eta_L^2 L^2}{2} + 15K^2\eta\eta_L^3 L^3\frac{m-n}{2n(m-1)})(\sigma_L^2 + 6K\sigma_G^2)\right]$ and $c$ is a constant. This completes the proof. $\quad\square$

### A.2.1 KEY LEMMAS

**Lemma 1** (Unbiased Sampling). *For strategies 1 and 2, the estimator $\Delta_t$ is unbiased, i.e.,*

$$\mathbb{E}_{S_t}[\Delta_t] = \bar{\Delta}_t.$$

*Proof of Lemma 1.*
Let $S_t = \{t_1, \cdots, t_n\}$ with size $n$. Both for sampling strategies 1 and 2, each sampling distribution is identical. Then we have:

$$\mathbb{E}_{S_t}[\Delta_t] = \frac{1}{n}\mathbb{E}_{S_t}\Big[\sum_{t_i \in S_t} \Delta_t^{t_i}\Big] = \frac{1}{n}\mathbb{E}_{S_t}\Big[\sum_{i=1}^{n} \Delta_t^{t_i}\Big] = \mathbb{E}_{S_t}[\Delta_t^{t_1}] = \frac{1}{m}\sum_{i=1}^{m} \Delta_t^i = \bar{\Delta}_t.$$

### A.3 AUXILIARY LEMMAS

**Lemma 2** (Lemma 4 in Reddi et al. (2020)). *For any step-size satisfying $\eta_L \leq \frac{1}{8LK}$, we can have the following results:*

$$\frac{1}{m}\sum_{i=1}^{m} \mathbb{E}[\|\mathbf{x}_{t,k}^i - \mathbf{x}_t\|^2] \leq 5K\eta_L^2(\sigma_L^2 + 6K\sigma_G^2) + 30K^2\eta_L^2\|\nabla f(\mathbf{x}_t)\|^2.$$

*Proof.* In order for this paper to be self-contained, we restate the proof of Lemma 4 in (Reddi et al., 2020) here.

For any worker $i \in [m]$ and $k \in [K]$, we have:

$\mathbb{E}[\|\mathbf{x}_{t,k}^i - \mathbf{x}_t\|^2] = \mathbb{E}[\|\mathbf{x}_{t,k-1}^i - \mathbf{x}_t - \eta_L g_{t,k-1}^t\|^2]$

$\leq \mathbb{E}[\|\mathbf{x}_{t,k-1}^i - \mathbf{x}_t - \eta_L(g_{t,k-1}^t - \nabla F_i(\mathbf{x}_{t,k-1}^i) + \nabla F_i(\mathbf{x}_{t,k-1}^i) - \nabla F_i(\mathbf{x}_t) + \nabla F_i(\mathbf{x}_t) - \nabla f(\mathbf{x}_t) + \nabla f(\mathbf{x}_t))\|^2]$

$\leq (1 + \frac{1}{2K-1})\mathbb{E}[\|\mathbf{x}_{t,k-1}^i - \mathbf{x}_t\|^2] + \mathbb{E}[\|\eta_L(g_{t,k-1}^t - \nabla F_i(\mathbf{x}_{t,k-1}^i))\|^2]$

$\quad + 6K\mathbb{E}[\|\eta_L(\nabla F_i(\mathbf{x}_{t,k-1}^i) - \nabla F_i(\mathbf{x}_t))\|^2] + 6K\mathbb{E}[\|\eta_L(\nabla F_i(\mathbf{x}_t) - \nabla f(\mathbf{x}_t))\|^2] + 6K\|\eta_L\nabla f(\mathbf{x}_t)\|^2$

$\leq (1 + \frac{1}{2K-1})\mathbb{E}[\|\mathbf{x}_{t,k-1}^i - \mathbf{x}_t\|^2] + \eta_L^2\sigma_L^2 + 6K\eta_L^2 L^2\mathbb{E}[\|\mathbf{x}_{t,k-1}^i - \mathbf{x}_t\|^2] + 6K\eta_L^2\sigma_G^2 + 6K\|\eta_L\nabla f(\mathbf{x}_t)\|^2$

$= (1 + \frac{1}{2K-1} + 6K\eta_L^2 L^2)\mathbb{E}[\|\mathbf{x}_{t,k-1}^i - \mathbf{x}_t\|^2] + \eta_L^2\sigma_L^2 + 6K\eta_L^2\sigma_G^2 + 6K\|\eta_L\nabla f(\mathbf{x}_t)\|^2$

$\leq (1 + \frac{1}{K-1})\mathbb{E}[\|\mathbf{x}_{t,k-1}^i - \mathbf{x}_t\|^2] + \eta_L^2\sigma_L^2 + 6K\eta_L^2\sigma_G^2 + 6K\|\eta_L\nabla f(\mathbf{x}_t)\|^2$

Unrolling the recursion, we get:

$$\frac{1}{m}\sum_{i=1}^{m} \mathbb{E}[\|\mathbf{x}_{t,k}^i - \mathbf{x}_t\|^2] \leq \sum_{p=0}^{k-1}(1 + \frac{1}{K-1})^p[\eta_L^2\sigma_L^2 + 6K\sigma_G^2 + 6K\eta_L^2\|\eta_L\nabla f(\mathbf{x}_t))\|^2]$$

$$\leq (K-1)[(1 + \frac{1}{K-1})^K - 1][\eta_L^2\sigma_L^2 + 6K\sigma_G^2 + 6K\eta_L^2\|\eta_L\nabla f(\mathbf{x}_t))\|^2]$$

$$\leq 5K\eta_L^2(\sigma_L^2 + 6K\sigma_G^2) + 30K^2\eta_L^2\|\nabla f(\mathbf{x}_t)\|^2$$

This completes the proof. $\quad\square$

# B   APPENDIX II: EXPERIMENTS

We provide the full detail of the experiments. We uses non-i.i.d. versions for MNIST and CIFAR-10, which are described as follows:

## B.1   MNIST

We study image classification of handwritten digits 0-9 in MNIST and modify the MNIST dataset to a non-i.i.d. version.

To impose statistical heterogeneity, we split the data based on the digits ($p$) they contain in their dataset. We distribute the data to $m = 100$ workers such that each worker contains only a certain class of digits with the same number of training/test samples. For example, for $p = 1$, each worker only has training/testing samples with one digit, which causes heterogeneity among different workers. For $p = 10$, each worker has samples with 10 digits, which is essentially i.i.d. case. In this way, we can use the digits in worker's local dataset to represent the non-i.i.d. degree qualitatively. In each communication round, 100 workers run $K$ epochs locally in parallel and then the server samples $n$ workers for aggregation and update. We make a grid-search experiments for the hyper-parameters as shown in Table 3.

Table 3: Hyper-parameters Tuning.

| | |
|---|---|
| Server Learning Rate | $\eta \in \{1, 10\}$ |
| Client Learning Rate | $\eta_L \in \{0.001, 0.01, 0.1\}$ |
| Local Epochs | $K \in \{1, 5, 10\}$ |
| Clients Partition Number | $n \in \{10, 50, 100\}$ |
| Non-i.i.d. Degree | $p \in \{1, 2, 5, 10\}$ |

We run three models: multinomial logistic regression, fully-connected network with two hidden layers (2NN) (two 200 neurons hidden layers with ReLU followed by an output layer), convolutional neural network (CNN), as shown in Table 4. The results are shown in Figures 2, 3 and 4.

Table 4: CNN Architecture for MNIST.

| Layer Type | Size |
|---|---|
| Convolution + ReLu | $5 \times 5 \times 32$ |
| Max Pooling | $2 \times 2$ |
| Convolution + ReLu | $5 \times 5 \times 64$ |
| Max Pooling | $2 \times 2$ |
| Fully Connected + ReLU | $1024 \times 512$ |
| Fully Connected | $512 \times 10$ |

## B.2   CIFAR-10

Unless stated otherwise, we use the following default parameter setting: the server learning rate and client learning rate are set to $\eta = 1.0$ and $\eta_L = 0.1$, respectively. The local epochs is set to $K = 10$. The total number of clients is set to 100, and the clients partition number is set to $n = 10$. We use the same strategy to distribute the data over clients as suggested in McMahan et al. (2016). For the i.i.d. setting, we evenly partition all the training data among all clients, i.e., each client observes 500 data; for the non-i.i.d. setting, we first sort the training data by label, then divide all the training data into 200 shards of size 250, and randomly assign two shards to each client. For the CIFAR-10 dataset, we train our classifier with the ResNet model. The results are shown in Figure 5 and Figure 6.

## B.3   DISCUSSION

**Impact of non-i.i.d. datasets:** Figure 2 shows the results of training loss (top) and test accuracy (bottom) for three models under different non-i.i.d. datasets with full and partial worker participation

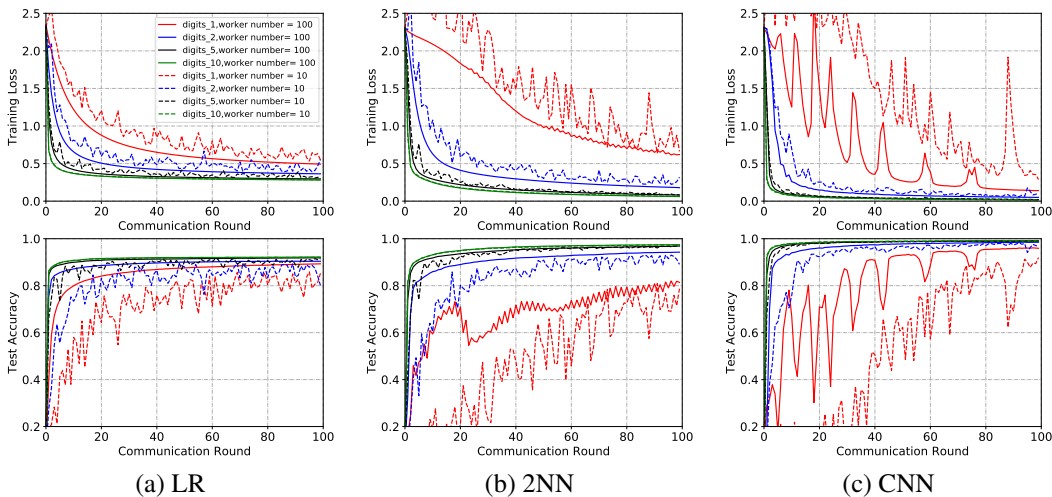

Figure 2: Training loss (top) and test accuracy (bottom) for three models on MNIST with hyperparameters setting: local learning rate 0.1, global learning rate 1.0, local steps 5 epochs.

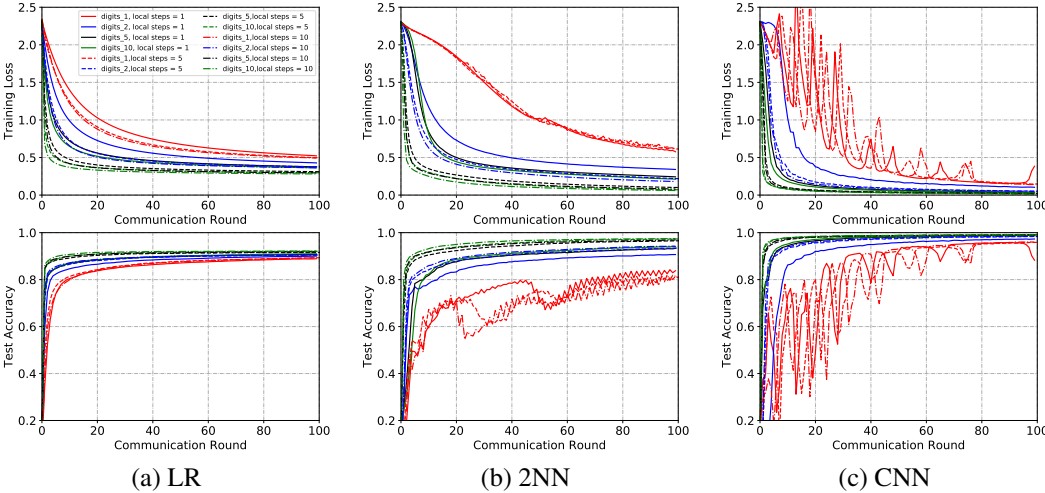

Figure 3: Training loss (top) and test accuracy (bottom) for three models on MNIST with hyperparameters setting: local learning rate 0.1, global learning rate 1.0, worker number 100.

on MNIST. We can see that the FedAvg algorithm converges under non-i.i.d. datasets with a proper learning rate choice in these cases. We believe that the major challenge in FL is the non-i.i.d. datasets. For these datasets with a lower degree of non-i.i.d., the FedAvg algorithm can achieve a good result compared with the i.i.d. case. For example, when the local dataset in each worker has five digits ($p = 5$) with full (partial) worker participation, the FedAvg algorithm achieves a convergence speed comparable with that of the i.i.d. case ($p = 10$). This result can be observed in Figure 2 for all three models. As the degree of non-i.i.d. datasets increases, its negative impact on the convergence is becoming more obvious. The higher the degree of non-i.i.d., the slower the convergence speed. As the non-i.i.d. degree increases (from case $p = 10$ to case $p = 1$), it is obvious that the training loss is increasing and test accuracy is decreasing. For these with high degree of non-i.i.d., the convergence curves oscillate and are highly unstable. This trend is more obvious for complex models such for CNN in Figure 2(c).

**Impact of worker number:** For full worker participation, the server can have an accurate estimation of the system heterogeneity after receiving the updates for all workers and neutralize this heterogeneity in each communication round. However, partial worker participation introduces another source of randomness, which leads to zigzagging convergence curves and slower convergence. In each

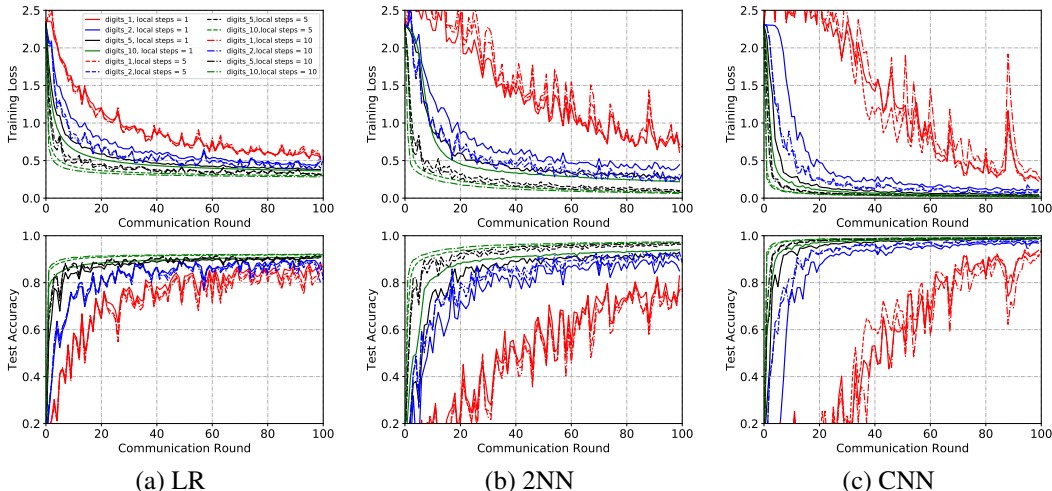

(a) LR                    (b) 2NN                    (c) CNN

Figure 4: Training loss (top) and test accuracy (bottom) for three models on MNIST with hyperparameters setting: local learning rate 0.1, global learning rate 1.0, worker number 10.

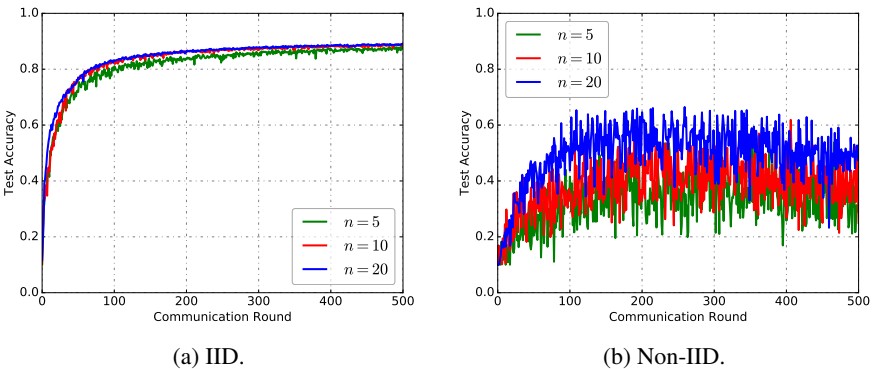

(a) IID.                    (b) Non-IID.

Figure 5: Test accuracy with respect to worker number on CIFAR-10 dataset.

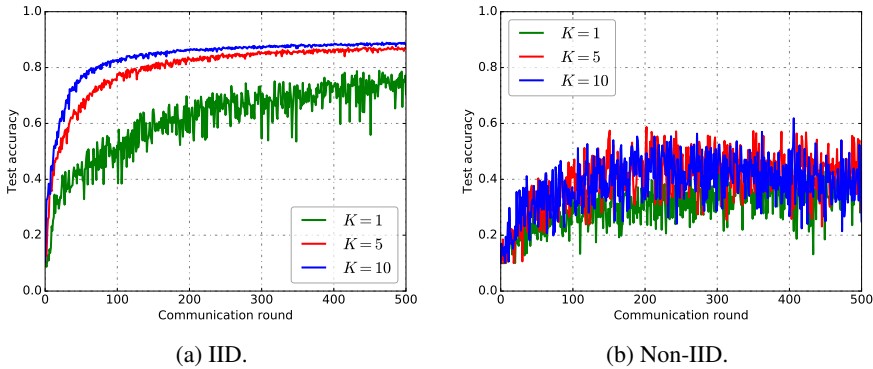

(a) IID.                    (b) Non-IID.

Figure 6: Test accuracy with respect to different local steps on CIFAR-10 dataset.

communication round, the server can only receive a subset of workers based on the sampling strategy. So the server could only have a coarse estimation of the system heterogeneity and might not be able to neutralize the heterogeneity among different workers for partial worker participation. This problem is more prominent for highly non-i.i.d. datasets. It is not unlikely that the digits in these datasets among all active workers are only a proper subset of the total 10 digits in the original MNIST dataset, especially with highly non-i.i.d. datasets. For example, for $p = 1$ with 10 workers in each

communication round, it is highly likely that the datasets formed by these ten workers only includes certain small number of digits (say, 4 or 5) rather than total 10 digits. But for $p = 5$, it is the opposite, that is, the digits in these datasets among these 10 workers are highly likely to be 10. So in each communication round, the server can mitigate system heterogeneity since it covers the training samples with all 10 digits. This trend is more obvious for complex models and datasets given the dramatic drop of test accuracy in the result of CIFAR-10 in Figure 5.

The sample strategy here is random sampling with equal probability without replacement. In practice, the workers need to be in certain states in order to be able to participate in FL (e.g., in charging or idle states, etc.(Eichner et al., 2019)). Therefore, care must be taken in sampling and enlisting workers in practice. We believe that the joint design of sampling schemes, number of workers and the FedAvg algorithm will have a significant impact on the convergence, which needs further investigations.

**Impact of local steps:** Figure 3 and Figure 4 shows the results of training loss (top) and test accuracy (bottom) for three models under different local steps with full and partial worker participation respectively. Figure 6 shows the impact of local steps in CIFAR-10. One open question of FL is that whether the local steps help the convergence or not. Li et al. (2019b) showed a convergence rate $\mathcal{O}(\frac{K}{T})$, i.e., the local steps may hurt the convergence for full and partial worker participation. In this two figures, we can see that local steps could help the convergence for both full and partial worker participation. However, it only has a slight effect on the convergence compared to the effects of non-i.i.d. datasets and number of workers.

**Comparison with SCAFFOLD:** We compare SCAFFOLD (Karimireddy et al., 2019) with the generalized FedAVg algorithm in this paper in terms of communication rounds, total communication overloads and estimated wall-clock time to achieve certain test accuracy in Table 2. We run the experiments using the same GPU (NVIDIA V100) to ensure the same conditions. Here, we give a specific comparison for these two algorithms under exact condition. Note that we divide the total training time to two parts: the computation time when the worker trains the local model and the communication time when information exchanges between the worker and server. We only compare the computation time and communication time with a fixed bandwidth $20MB/s$ for both uploading and downloading connections. As shown in Figure 7, to achieve $\epsilon = 75\%$, SCAFFOLD performs less communication round due to the variance reduction techniques. That is, it spends less time on computation. However, it needs to communicates as twice as the FedAvg since the control variate to perform variance reduction in each worker needs to update in each round. In this way, the communication time would be largely prolonged.

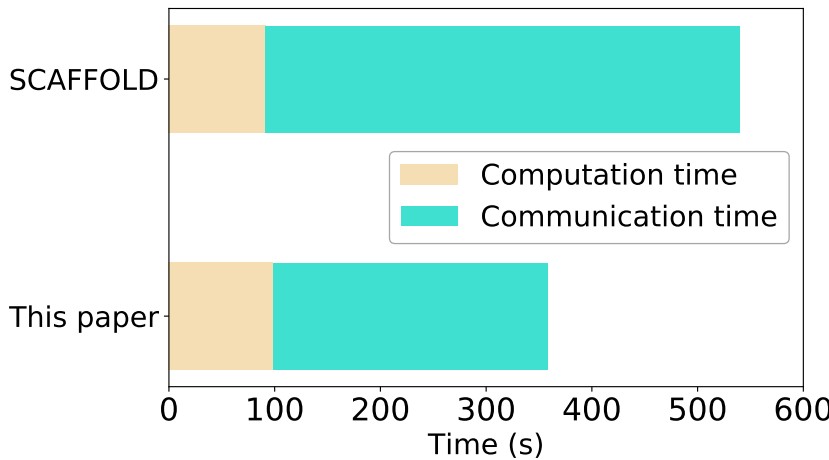

Figure 7: Wall-clock time to achieve test accuracy $\epsilon = 75\%$ on CIFAR-10 dataset.

