# OpenReview forum: "Achieving Linear Speedup with Partial Worker Participation in Non-IID Federated Learning"
_ICLR.cc/2021/Conference — ICLR 2021 Poster_

### Official Review · AnonReviewer4 · 2020-10-28
**Good theoretical results, though the model for the partial worker participation is too strong**

**Rating:** 7
**Confidence:** 5

**Review:**

This paper provides a new analysis for the FedAvg algorithm, which assumes the data on different workers are non-IID and the objective functions are non-convex. The new analysis improved the existing bounds of FedAvg. Besides, the analysis is also extended to the non-stationary network, where the number of workers participating in the optimization may vary.

As FedAvg is the most important optimization algorithm in federated learning, I think such theoretical progress is good.

If my understanding is correct, as the new bound allows a larger number of local steps, one important improvement for the algorithm is that the number of communication rounds can be also reduced. I suggest the authors demonstrate this fact in the paper, e.g., by adding a column concerning communication complexity in Table 1 and appending some discussion about communication cost.

One drawback of the theoretical results is that the model for the non-stationary network is too strong. It assumes the number of participants in each round is fixed, and each worker has an equal probability to participate. Such an assumption can be rarely satisfied in practice.

One typo: I guess $p_i=\frac{1}{m}$ should be $p_i=\frac{n}{m}$ in the last line of page 5.

---

> ### Author Response · Authors · 2020-11-18
> **Response to Reviewer 4's Comments**
>
> Thank you very much for your review and constructive comments, which helped us significantly improve the quality of this paper. In this revision, we have carefully revised our paper based on your comments, questions, and suggestions. The re-written parts are highlighted in blue. Please see the revised manuscript at the top of this page.
> The detailed point-by-point responses are as follows:
>
> > **Your comment:** 1. If my understanding is correct, as the new bound allows a larger number of local steps, one important improvement for the algorithm is that the number of communication rounds can be also reduced. I suggest the authors demonstrate this fact in the paper, e.g., by adding a column concerning communication complexity in Table 1 and appending some discussion about communication cost.
>
> **Our response:**
> Thanks for your suggestion.
> Following your suggestion, in this revision, we have rewritten and added a new column for communication cost comparisons in term of the reduced communication round in Table 1 (cf. Page 3).
> We have also added detailed discussions on communication cost comparisons in the contribution on Page 2 and Remark 3 on Page 6.
>
> > **Your comment:** 2. One drawback of the theoretical results is that the model for the non-stationary network is too strong. It assumes the number of participants in each round is fixed, and each worker has an equal probability to participate. Such an assumption can be rarely satisfied in practice.
>
> **Our response:**
> Thanks for your insightful comments.
> We agree that the requirement $|S_t| = n$ could be a restrictive condition in practice.
> But we note that this condition is not critical in our algorithm design and can be relaxed.
> We adopted the $|S_t| = n$ simply for simplicity and ease of presentation in our initial submission.
> In this revision, we showed that the requirement $|S_t| = n$ can be relaxed to $|S_t| \geq n$.
> That is, the same convergence rate can be guaranteed as long as there are at least $n$ workers in each communication round (no need to be exactly $n$).
> Specifically, we have added Remark 7 for this issue (cf. Page 7) and revised the proofs accordingly (cf. Pages 16 and 19) to highlight this relaxed requirement on $S_t$.
>
> > **Your comment:** 3. One typo: I guess $p_i = \frac{1}{m}$ should be $p_i = \frac{n}{m}$ in the last line of page 5.
>
> **Our response:**
> Thanks for your question.
> There is no typo here and this question is likely caused by our somewhat ambiguous writing and the $p_i$ notation.
> To avoid this confusion, we have rewritten the sentence in the last line of Page 5 in our initial submission (now on Page 6 in this revision).
> Specifically, for each member in the participation set $S_t$, we pick a worker from the entire set $[m]$ uniformly at random with probability $p_i = \frac{1}{m}$.
> As a result, the selection likelihood for each worker in $S_t$ is $\mathbb{P}[i \in S_t] = \frac{n}{m}$.
> Therefore, your understanding is also correct.
> We hope this clarifies your confusion.

---

> > ### Comment · AnonReviewer4 · 2020-11-25
> > **I have read the response and decide to keep the score unchanged.**
> >
> > I think the current score 7 is still appropriate after reading the authors' response.

---

### Official Review · AnonReviewer1 · 2020-10-28
**Official Blind Review #1**

**Rating:** 6
**Confidence:** 5

**Review:**

EDIT after discussion:

Thanks for the authors' comments and revisions to this paper. I'm glad to see that some of my concerns have been addressed so I decide to change my score.

Just some minor comments about writing for the revised version. I think the proof sketch is clear and easy to follow the logic with some room for further improvement. Like the other reviers' comments, I think some part of the paper like notations should be with clear explanation.

-------------------------------------------------------------

This paper provides a theoretical analysis for FedAvg, the seminal optimization algorithm for federated learning. The analysis is featured at client sampling and non-iid distributions among clients, which are two well-known challenges in federated learning. The theoretical results in this paper provide us an optimization view to understand why FedAvg works, and how it saves communication cost by allowing low SGD updates and by model averaging.

I think the authors can further improve this paper from the following aspects.

First, the metric for convergence rate is slightly less common in the methods shown in Table 1. Most existing works (e.g., Yu et al. 2019a, Karimireddy et al. 2019) use the average norm of stochastic gradients in the previous T steps. However in this paper the authors consider the minimum of these T values. It's not a problem per se because the minimum also shows the convergence trend somehow. The problem is, comparison with existing work as listed in Table 1 looks confusing to me. I would like to see if there is a way of showing the linear speedup in the same fashion.

Second, I think the experiment section has some room for improvement. The whole section doesn't show comparison with any existing but some curves with hyperparameter variants. In particular, I really want to see how the proposed algorithm compared with SCAFFOLD, because both rates are matched (despite my comment above).

Like the authors said, SCAFFOLD needs to maintain an additional set of variables for variance reduction, which may introduce more communication and computation costs. For this reason, I would like to see comparison not only the convergence w.r.t. steps/epochs, but also in wall clock, under the same bandwidth constraint, FLOPS, .etc.

One minor suggestion is about writing. The technique to show the linear speedup in this paper is worth writing some lines of proof sketch. One such example is the SCAFFOLD paper. It will help readers to understand the main contribution, and may help readers to apply such technique in other related problems.

---

> ### Author Response · Authors · 2020-11-18
> **Response to Reviewer 1's Comments (1/2)**
>
> Thank you very much for your review and constructive comments, which helped us significantly improve the quality of this paper. In this revision, we have carefully revised our paper based on your comments, questions, and suggestions. The re-written parts are highlighted in blue. Please see the revised manuscript at the top of this page. The detailed point-by-point responses are as follows:
>
> > **Your comment** 1. First, the metric for convergence rate is slightly less common in the methods shown in Table 1. Most existing works (e.g., Yu et al. 2019a, Karimireddy et al. 2019) use the average norm of stochastic gradients in the previous T steps. However in this paper the authors consider the minimum of these T values. It's not a problem per se because the minimum also shows the convergence trend somehow. The problem is, comparison with existing work as listed in Table 1 looks confusing to me. I would like to see if there is a way of showing the linear speedup in the same fashion.
>
> **Our response:** Thanks for your insightful comments.
> As you mentioned, both metrics are valid for characterizing convergence rates. Moreover, it is worth pointing out the use of $\min_{t \in [T]} (\cdot)$ or $(1/T) \sum_{t=1}^{T} (\cdot)$ as the metric for convergence analysis does not change our convergence rate results in this paper. To see this, we note that the common approach for non-convex convergence analysis is to first derive the one-step descent bound, then compute the telescoping sum from $1$ to $T$, and finally average on $T$ to obtain the convergence rate bound measured by $\frac{1}{T} \sum_{t=1}^{T} \mathbb{E} || \nabla f(x_t) ||^2_2$. Most of the related works adopt this approach and so does our paper. Therefore, our convergence rate bounds are the same under $\frac{1}{T} \sum_{t=1}^{T} \mathbb{E} || \nabla f(x_t) ||^2_2$ (cf. Pages 13, 16 and 19), i.e., the average norm of gradients. Also, from the basic fact that $\min_{t \in [T]} \mathbb{E} ||\nabla f(x_t) ||^2_2 \leq \frac{1}{T} \sum_{t=1}^{T} \mathbb{E} || \nabla f(x_t) || ^2_2$, it follows that the *same* convergence rate bound in our paper also holds for $\min_{t \in [T]} \mathbb{E} ||\nabla f(x_t) ||^2_2$. Furthermore, since the proof strategies are the same in all related works, the convergence rate results in Table 1 are directly comparable, although they are defined somewhat differently.
>
> Note that it is also not uncommon to use $\min_{t \in [T]} (\cdot)$ as the metric in the  literature, e.g., in FL (Reddi et al., 2020) and in other non-convex convergence analysis [1][2] below. Since the use of the convergence metric can be one way or the other among these two under the same analysis approach, we adopt $\min_{t \in [T]} (\cdot)$ or $\frac{1}{T} \sum_{t=1}^{T} (\cdot)$ without any significant theoretical implications. Again, we emphasize that the convergence rate bounds of our work in Table 1 are exactly the same under both convergence metrics and are comparable to related works.
>
> [1]. Ward, Rachel, Xiaoxia Wu, and Leon Bottou. AdaGrad Stepsizes: Sharp Convergence over Nonconvex Landscapes, In International Conference on Machine Learning, pp. 6677-6686. 2019.
>
> [2]. Chen, Xiangyi, Sijia Liu, Ruoyu Sun, and Mingyi Hong. On the Convergence of A Class of Adam-Type Algorithms for Non-Convex Optimization, In International Conference on Learning Representations. 2018.

---

> > ### Author Response · Authors · 2020-11-18
> > **Response to Reviewer 1's Comments (2/2)**
> >
> > > **Your comment:** 2. Second, I think the experiment section has some room for improvement. The whole section doesn't show comparison with any existing but some curves with hyperparameter variants. In particular, I really want to see how the proposed algorithm compared with SCAFFOLD, because both rates are matched (despite my comment above).
> > >Like the authors said, SCAFFOLD needs to maintain an additional set of variables for variance reduction, which may introduce more communication and computation costs. For this reason, I would like to see comparison not only the convergence w.r.t. steps/epochs, but also in wall clock, under the same bandwidth constraint, FLOPS, .etc.
> >
> > **Our response:** Thanks for your suggestions.
> > As you suggested, in this revision, we have added extensive experiments to compare with SCAFFOLD.
> > Under the same condition (such as uplink and downlink bandwidths, FLOPS, etc.), we re-run three models (Logistic Regression, 2NN and CNN) on MNIST and the Resnet-18 model on CIFAR-10 with i.i.d. and non-i.i.d. settings.
> > We summarized all results in Table 2 with associated explanations on Page 9, and added an extra figure in appendix (cf. Page 24).
> > We provide a quick summary of the main results here:
> > * For simple models on MNIST, both algorithms have similar converges speeds.
> > * For large models on CIFAR-10 with both i.i.d. and non-i.i.d. datasets, although our FedAvg algorithm takes more communication rounds to achieve a test accuracy $\epsilon=0.75$, the total training time of our FedAvg algorithm is shorter than that of SCAFFOLD.
> > * The SCAFFOLD algorithm needs to take more than 1.5 times of communication load and walk-clock time compared to those of our FedAvg algorithm due to the extra information exchange in order to perform variance reduction.
> > As a result, the communication time is significantly longer, which leads to longer overall training time.
> > For example, for Resnet-18 model on non-i.i.d. CIFAR-10 to achieve $\epsilon=0.75$ test accuracy, our FedAvg algorithm takes 61 communication rounds and requires $5200.29$ MB communication cost.
> > In comparison, the SCAFFOLD algorithm takes 52 communication rounds but requires $8866.06$ MB communication cost.
> > This results in longer total training time for SCAFFOLD.
> >
> > > **Your comment:** 3. One minor suggestion is about writing. The technique to show the linear speedup in this paper is worth writing some lines of proof sketch. One such example is the SCAFFOLD paper. It will help readers to understand the main contribution, and may help readers to apply such technique in other related problems.
> >
> > **Our response:**
> > Thanks for your suggestion.
> > As you suggested, in this revision, we have added two proof sketches immediately after Theorems 1 and 2 to make it easier to understand the main strategies and techniques used in the proofs (cf. Pages 5, 6).

---

### Official Review · AnonReviewer3 · 2020-10-28
**In overall, I believe this is a decent contribution to the ICLR community which gives a clear analysis of the federated averaging in more realistic scenarios such as non I.I.D data accross devices and partial device participation. Some improvements are needed in writing and there are some concerns about the applicability to large number of workers that needs to be addressed.**

**Rating:** 7
**Confidence:** 4

**Review:**

#### Summary of the paper:
This paper analyzes convergence rates of the standard "FedAvg" algorithm  (with two sided learning rates) that is widely used in federated learning in multiple orthogonal perspectives:
a) non I.I.D datasets - with reasonable assumptions about the heterogeneity and suitable parameters, convergence guarantees comparable to I.I.D setting can be achieved.
b) partial worker participation - provides an analysis of convergence based on the number of workers participated (constant $n$) in each round.
c) number of local steps - analysis of overall convergence rate that depends on this is provided, but full effects are not yet clear (open problem).
This paper provides ample empirical evaluations to validate the theoretical claims shown.

#### Quality:
This paper is generally well written and structured well. Although I did not read the proof of the theorems thoroughly, the claims sound natural and believable. Thorough explanations and insights of the theoretical claims are provided.

#### Clarity:
Clarity of certain parts needs to be improved.
- The claim about the number of local updates $K$ improving to $T/m$ from $T^{1/3}/m$ is not clear. At first glance this does not look like an improvement because it gets worse for the the workers(devices). But remark 3 explains that this indeed causes an improvement in the overall convergence rate. I think this should be clearly mentioned in the contributions.
- It is better if the authors can change some superscripts for footnotes in table 1, the superscripts 4 and 6 look like exponents at the first glance which can be confusing.
- In related work, it is mentioned that "(Karimireddy et al., 2019) can achieve linear speedup but extra variance reduction operations are required, which lead to high communication costs and implementation complexity. In this paper, we show that this linear speedup can be achieved without any extra requirements." State this clearly in the contributions since the table 1 does not imply this and it is difficult to understand the improvements, thus the novelty of these results by looking at table 1 or contributions.
-  I believe the the expectations in assumption 2 and theorems 1, 2 are over local datasets samples. Clearly state this.
- $f_0, f_*$ are not defined in theorem 1,2 (I believe they are loss function values at start and the optimal point). Define these quantities.


#### Originality and significance:
Although this paper does not technically introduce a new algorithm for federated, I believe that the theoretical analysis of "FedAvg" in the non I.I.D. and partial worker participation setting and relevant parameters are nice contributions to the ICLR community. I think the insights obtained by this work can be very useful in developing better algorithms and in practice since it considers more realistic scenarios and the model assumptions are reasonable.

Pros:
- Main contributions on non I.I.D data and partial worker settings are nice and relevant realistic scenarios.
- I think the result on linear speed up that does not depend on a requirement of bounded gradient assumption is nice especially in non I.I.D and non-convex settings since such dependencies in these scenarios can be unrealistic.
- Extensive experimental evaluations on the theoretical claims validate the results.

Cons:
- One concern I have is the claimed convergence rate in FedAvg is achievable when $T \ge mK$, but in federated learning settings, it is  often the case that $m$ is very large. Therefore this requires $T$ to be very large and thus blow up the required number of communication rounds for convergence. Is this a common issue in the other works compared in table 1 or inherent in FedAvg? The experiments are also done with $m=100$, therefore it is difficult to gauge the impact of large(which is realistic) number of clients in practice.

Other comments:
I would like to understand that the requirement $|S_t| = n$ in section 3.2 can be relaxed to something like at least $n$ workers in each round and the same convergence guarantees would hold. While strategy 1 and 2 are viable, I believe this can also be a nice way to understand the partial worker setting.

Final feedback:
I thinks this paper has decent contributions to the understanding of federated averaging method in more realistic scenarios. I am willing to increase my score if the concerns mentioned above are properly addressed.
===============================================================================================================

Added after reading author response:

------------------------------------------------------

I believe authors have addressed the issues I raised about clarity in certain parts of the paper in their response sufficiently.  I agree that in this paper, the convergence rates' dependency on $m$ has been improved significantly compared to previous work. Thus I increase my score. I encourage authors to investigate $m$'s effects on the convergence rate more to see whether there is a structural limitation in federated learning settings, perhaps better left for future work.

---

> ### Author Response · Authors · 2020-11-18
> **Response to Reviewer 3's Comments (1/2)**
>
> Thank you very much for your review and constructive comments, which helped us significantly improve the quality of this paper. In this revision, we have carefully revised our paper based on your comments, questions, and suggestions. The re-written parts are highlighted in blue. Please see the revised manuscript at the top of this page.
> The detailed point-by-point responses are as follows:
>
> > **Your comment:** 1. Clarity: Clarity of certain parts needs to be improved.
> >* The claim about the number of local updates $K$ improving to $T/m$ from $T^{1/3}/m$ is not clear. At first glance this does not look like an improvement because it gets worse for the the workers(devices). But remark 3 explains that this indeed causes an improvement in the overall convergence rate. I think this should be clearly mentioned in the contributions.
> >* It is better if the authors can change some superscripts for footnotes in table 1, the superscripts 4 and 6 look like exponents at the first glance which can be confusing.
> >* In related work, it is mentioned that "(Karimireddy et al., 2019) can achieve linear speedup but extra variance reduction operations are required, which lead to high communication costs and implementation complexity. In this paper, we show that this linear speedup can be achieved without any extra requirements." State this clearly in the contributions since the table 1 does not imply this and it is difficult to understand the improvements, thus the novelty of these results by looking at table 1 or contributions.
> >* I believe the the expectations in assumption 2 and theorems 1, 2 are over local datasets samples. Clearly state this.
> >* $f_0, f_*$ are not defined in theorem 1,2 (I believe they are loss function values at start and the optimal point). Define these quantities.
>
> **Our response:** Thanks for pointing out these clarity issues. In this revision, we provide further clarifications as follows:
> * Your understanding is correct: the larger number of local steps $K$ does lead to improved convergence rate as pointed out by Corollary 1 and Remark 3, which is a new insight not found in the current literature and a contribution of this work. In this work, we have clarified the meaning of local updates $K$'s improvement and added this into the contributions in Introduction (cf. Page 2, the second bullets under contributions).
>
> * Thanks for pointing out this confusion. In this revision, we have moved the locations of superscripts for footnotes in Table 1 to avoid the confusion (cf. Page 3, Table 1).
>
> * Thanks for your suggestions. As you suggested, in this revision, we have added discussions of the results in (Karimireddy et al., 2019) in the contribution to highlight improvements (cf. Page 2). Moreover, we have added a new column in Table 1 to compare the communication complexities of the related work and this paper. We hope that, with these two explicit additions, the communication improvements of our work is made clear.
>
> * Yes, your understanding is correct. The expectations are indeed over local dataset samples. We have added this clarification in Assumption 2, Theorems 1 and 2 (cf. Pages 4, 5, 6).
>
> * Thanks for pointing out these undefined notation. In this revision, we have added the definitions of $f_0$ and $f_{*}$ in Theorems 1 and 2 (cf. Pages 5, 6).

---

> > ### Author Response · Authors · 2020-11-18
> > **Response to Reviewer 3's Comments (2/2)**
> >
> > > **Your comment:** 2. Cons: One concern I have is the claimed convergence rate in FedAvg is achievable when $T>mk$, but in federated learning settings, it is often the case that $m$ is very large. Therefore this requires $T$ to be very large and thus blow up the required number of communication rounds for convergence. Is this a common issue in the other works compared in table 1 or inherent in FedAvg? The experiments are also done with $m=100$, therefore it is difficult to gauge the impact of large(which is realistic) number of clients in practice.
> >
> > **Our response:** Thanks for your insightful comments.
> > Yes, the required large value of $T$ is a common issue for all related work compared in Table 1. Specifically, in order to achieve linear speedup, it is required that $T$ should be relatively large, so that the first $1/T$-dependent term in the convergence rates of all algorithms becomes negligible compared to the second $1/\sqrt{T}$-dependent term (hence, the overall convergence rate is approximately $O(1\sqrt{mKT})$, i.e., *a linear speedup*. In other words, the notion of *linear speedup* is relevant for all these FL algorithms only when $T$ is sufficiently large. Nonetheless, one of our key contributions in this work is that we improve the value of $T$ by *two orders of magnitude* from $\Omega(m^3)$ to $\Omega(m)$. To see this, note that the two state-of-art bounds in previous works are $T \geq (mK)^3$ (Karimireddy et al., 2019; Yu et al., 2019a; Kairouz et al., 2019) and $T\geq m^3K$ (Wang and Joshi (2018)). In contrast, our lower bound for linear speedup is $T \geq mK$.
> >
> > As for FL in practice, it is reasonable to set $m \sim 100$ in cases such as the Cross-silo FL in Kairouz et al., (2019).
> > Also, the $m = 100$ setting has been used in numerical experiments in quite a few existing works (e.g., McMahan et al., 2016; Karimireddy et al., 2019;  Li et al., 2019b; Sattler et al., 2019). For fair comparisons with these prior works and also due to computation resource limitation, we adopt $m=100$ in this paper. Note that there also exist other numerical settings with even smaller or the same level $m$-values in the literature, such as $m = 10$ (Zhao et al., 2018), $m = 32$ (Wang et al., 2019b), $m = 500$ (Reddi et al., 2020) and so on. Meanwhile, some industrial-scale FL systems (e.g., Google Gboard and examples in Kairouz et al., (2019)) could have $m$-values up to the order of $m \sim 10^{10}$. But this is beyond our experiment capacity.
> >
> > >**Your comment:** 3. Other comments: I would like to understand that the requirement in section 3.2 can be relaxed to something like at least  workers in each round and the same convergence guarantees would hold. While strategy 1 and 2 are viable, I believe this can also be a nice way to understand the partial worker setting.
> >
> > **Our response:** Thanks for this insightful comment. Yes, the requirement $|S_t| = n$ can indeed be relaxed to $|S_t| \geq n$.
> > That is, the same convergence rate can be guaranteed if at least $n$ workers in each communication round (no need to be exactly $n$). In this revision, we have added Remark 7 to clarify this issue (cf. Page 7) and revised the proofs accordingly (cf. Pages 16 and 19) to highlight this relaxed requirement on $S_n$. We agree with the reviewer that this relaxation offers a deeper understand on the requirement of partial worker participation.

---

### Decision · Program_Chairs · 2021-01-07
**Final Decision**

**Decision:**

Accept (Poster)

**Comment:**

The authors have addressed the issues raised by the reviewers. All the reviewers think that the paper deserves to be published at ICLR 2021. The authors should implement all the reviewers’ suggestions into the final version, especially for clarity issues and clear explanations. The reviewer also encourages authors to investigate $m$’s effects on the convergence rate more to see whether there is a structural limitation in federated learning settings for future work.